# Invasive Californian death caps develop mushrooms unisexually and bisexually

Yen-Wen Wang [1] ✉, Megan C. McKeon [2,3], Holly Elmore[4], Jaqueline Hess[5], Jacob Golan[1], Hunter Gage[3], William Mao[1], Evan Harrow[1,6], Susana C. Gonçalves [7], Christina M. Hull[3,8] & Anne Pringle [1,9]

Canonical sexual reproduction among basidiomycete fungi involves the fusion of two haploid individuals of different mating types, resulting in a hetero-karyotic mycelial body made up of genetically different nuclei. Using population genomics data and experiments, we discover mushrooms of the invasive and deadly *Amanita phalloides* can also be homokaryotic; evidence of sexual reproduction by single, unmated individuals. In California, genotypes of homokaryotic mushrooms are also found in heterokaryotic mushrooms, implying nuclei of homokaryotic mycelia are also involved in outcrossing. We find death cap mating is controlled by a single mating type locus, but the development of homokaryotic mushrooms appears to bypass mating type gene control. Ultimately, sporulation is enabled by nuclei able to reproduce alone as well as with others, and nuclei competent for both unisexuality and bisexuality have persisted in invaded habitats for at least 17 but potentially as long as 30 years. The diverse reproductive strategies of invasive death caps are likely facilitating its rapid spread, suggesting a profound similarity between plant, animal and fungal invasions.

Invasion biology focuses on plants and animals and their diseases, while the changing geographical distributions of other microbes go largely unnoticed[1]. The poisonous, European *Amanita phalloides* (the death cap) is an ectomycorrhizal agaricomycete fungus introduced to North America[2,3]. Death caps are now abundant along the California coast and each year cause human and animal fatalities[4,5]. The mechanisms driving the spread of the fungus are not understood. In plants and animals, successful spread is associated with possession of multiple reproductive strategies: After an introduction, the ability to propagate vegetatively and sexually, and to sexually reproduce with-out a mate, are advantageous[6,7]. We sought to understand how death caps reproduce, and whether selfing is a strategy used by *A. phalloides* to sporulate and move across landscapes.

Agaricomycete fungi are characterized by bisexual (heterothallic) reproduction[8]: To complete the life cycle, two haploid mycelia of different mating types fuse and form a heterokaryotic, functionally diploid mycelium. Each cell of the mycelium houses two genetically different haploid nuclei. Mushrooms (sporocarps) develop from the heterokaryotic mycelium, and like the mycelium, carry two nuclei per cell. Within a mushroom's gills or pores, nuclei briefly fuse before undergoing meiosis to produce sexual basidiospores[9]. By contrast, unisexual fungi reproduce in the absence of a second mycelium[10–12]. The resulting mushrooms are homokaryotic and so lack heterozygosity[9]. In the laboratory, bisexual agaricomycetes can be forced to develop sporocarps from one haploid mycelium with a single mating type in response to physical, chemical, or genetic

[1]Department of Botany, University of Wisconsin-Madison, Madison, WI, USA. [2]Department of Genetics, University of Wisconsin-Madison, Madison, WI, USA. [3]Department of Biomolecular Chemistry, University of Wisconsin-Madison, Madison, WI, USA. [4]Rethink Priorities, San Francisco, CA, USA. [5]Cambrium GmbH, Berlin, Germany. [6]Department of Chemistry, University of Wisconsin-Madison, Madison, WI, USA. [7]Centre for Functional Ecology, Department of Life Sciences, University of Coimbra, Coimbra, Portugal. [8]Department of Medical Microbiology & Immunology, University of Wisconsin-Madison, Madison, WI, USA. [9]Department of Bacteriology, University of Wisconsin-Madison, Madison, WI, USA. ✉e-mail: yen-wen.wang@yale.edu

manipulations[13,14]. However, mushrooms forced from haploid mycelia grown in laboratory environments are typically abnormal[15]. Wild populations able to generate functional sporocarps both bisexually and unisexually in nature are unknown.

Here, we show invasive death caps in California can develop mushrooms and sporulate both bisexually and unisexually. The nuclei of unisexual mushrooms are also found with genetically different nuclei in bisexual mushrooms; these nuclei reproduce alone and with others. Death caps often possess unusual numbers of spores on each spore-bearing structure (basidium). While mating is controlled by a single mating type locus, reproduction by unisexual individuals appears to bypass the control of mating type determining genes. Finally, unisexual nuclei have persisted in invaded habitats for decades, and they span large territories.

## Results and discussion
### Invasive, homokaryotic mushrooms
To elucidate the reproductive strategies used by invasive *A. phalloides*, we sequenced genomes of 86 mushrooms collected from Point Reyes National Seashore (PRNS) in California between 1993 and 2015 ($N = 67$), from three sites in Portugal in 2015 ($N = 11$), and from other European countries between 1978 and 2006 ($N = 8$) (Supplementary Data 1). As a fungus grows in a habitat, a single mycelium can develop one or multiple sporocarps, and the 86 mushrooms resolve into 37 distinct genetic individuals (Supplementary Fig. 1a and Supplementary Data 1). Unexpectedly, the estimated heterozygosities of two Californian individuals, g21 and g22, were ten times lower than heterozygosities of other individuals (Fig. 1a and Supplementary Note 1). Individual g21 was collected as two mushrooms in 2014 and g22 as six mushrooms in 2004 and 2014. We hypothesized these two Californian individuals are homokaryotic (see also Supplementary Discussion). To test the hypothesis, we used raw read data to identify unique, short DNA strings (unique k-mers) and quantified the number of appearances of each unique k-mer (each k-mer's depth). Heterozygotic genomes normally show two peaks in k-mer depth, a primary peak and a secondary peak with half the depth of the primary peak (Fig. 1b). The secondary peak is generated by the heterozygous SNPs within a genome and was absent for our putatively homokaryotic individuals (Fig. 1b). In parallel, we investigated sequencing frequencies of alleles at putatively heterozygous sites. Because a true heterozygous site is made up of two alleles, the frequency for each heterozygous allele in sequencing reads should center at 50%, with deviations caused by stochasticity. Once again, putatively homokaryotic individuals are different; the frequency spectra of g21 and g22 are flat (Fig. 1c).

To determine if homokaryotic individuals can mate with other *A. phalloides* or are reproductively isolated, we estimated kinship among the homokaryotic and other heterokaryotic individuals using an algorithm developed by us for use in organisms with mixed haploid/diploid genetic systems, enabling us to identify heterokaryotic individuals housing nuclei from a homokaryon[16] (Supplementary Note 2). Many heterokaryotic individuals house either the g21 or g22 nucleus. We cannot distinguish whether these individuals are the parents or offspring of the homokaryons, nonetheless, because we identified more than one heterokaryotic individual as either the parent or offspring of both g21 and g22, and because only one individual can function as a parent of a homokaryon, we can identify other individuals as offspring. Thus, homokaryotic individuals appear to mate with other individuals (Fig. 2 and Supplementary Fig. 1b, c). To test whether homokaryotic individuals represent diverged lineages (e.g., cryptic species), we built gene trees from the genomes using 3324 universal single-copy orthologs (BUSCOs) and combined the gene trees using coalescent-based methods. In the combined tree, homokaryons were neither sister groups nor diverged lineages of the heterokaryotic *A. phalloides* (Supplementary Fig. 2); thus, homokaryotic individuals are not reproductively isolated.

### Unusual spore numbers
While laboratory sporocarps generated from cultured haploid mycelia are typically aberrant[14,15], the homokaryotic sporocarps we collected in nature were not very different from the heterokaryotic sporocarps collected from the same sites. In 2021 we revisited PRNS and again collected homokaryotic sporocarps generated by individuals g21 and g22, this time confirming genetic identity by Sanger sequencing of 11 loci (Supplementary Tables 1, 2). Collected sporocarps are morphologically similar to heterokaryotic sporocarps (Fig. 3a, b). Using dried materials from the original collections made in 2014, we discovered that both homokaryotic and heterokaryotic sporocarps possess unisporic, bisporic, and trisporic basidia, as well as canonical tetrasporic basidia (Fig. 3c, d and Supplementary Fig. 3). However, the ratios of the spore arrangements were different among the five individuals we measured (Fig. 3d), with heterokaryotic g25 possessing the highest frequency of tetrasporic basidia and homokaryotic g21 possessing the highest frequency of unisporic basidia. Next, we imaged patterns of nuclei within basidia and basidiospores. As documented in closely related *Amanita* species[17], younger basidiospores house one nucleus, and more mature spores house two, likely the result of a mitotic division within developing basidiospores (Supplementary Fig. 4a, b). Tetrasporic basidia leave no nuclei behind in the originating basidium (Supplementary Fig. 4a, b), but in trisporic basidia, one nucleus remains in the basidium (Fig. 3e, f and Supplementary Fig. 4c). The number of spores on a basidium does not appear to influence nuclear segregation; regardless of spore number, each spore receives one meiotic nucleus from the originating basidium, a phenomenon also observed in bisporic *A. bisporigera*[18]. Pseudohomothallism, or the phenomenon of a spore with two genetically different nuclei, and hence, the ability to grow and sexually reproduce without a mate[12], is not a feature of the *A. phalloides* life cycle. The same spore-nuclear dynamics are observed in both heterokaryotic and homokaryotic sporocarps, suggesting basidia of both kinds of sporocarps are cytologically similar. Moreover, fluorescent staining of nuclei in mycelia taken from stipe tissues revealed *A. phalloides* does not possess clamp connections and cells of both homokaryotic and heterokaryotic sporocarps can be multinucleate (Supplementary Fig. 4d, e).

### Genetics of sexual reproduction
Because sporocarp development is closely associated with sexual reproduction, we next sought to determine the genetics of mating system within the genus *Amanita*. Canonical agaricomycete mating systems are controlled by two mating (MAT) loci: a pheromone and

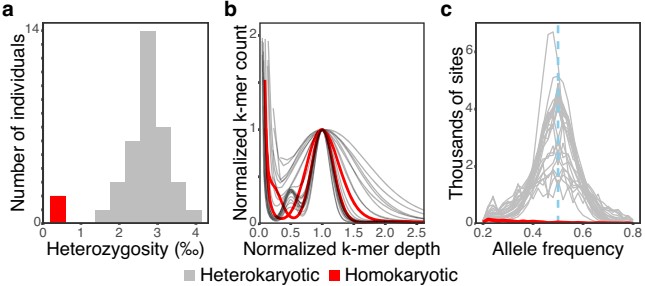

**Fig. 1 | Genomes of two putatively homokaryotic individuals collected in California bear signatures of a haploid or homozygotic genome. a** Whole-genome heterozygosities of 37 individuals; note the cluster of two Californian individuals at left (in red). **b** Peaks of k-mer depths for Californian and Portuguese individuals; a secondary peak at 0.5 implies heterozygosity and is lacking for the two Californian individuals (in red). **c** Sequencing frequencies of variable SNPs within individuals; peaks at 0.5 indicate heterozygosity and are lacking for the two putative homokaryons.

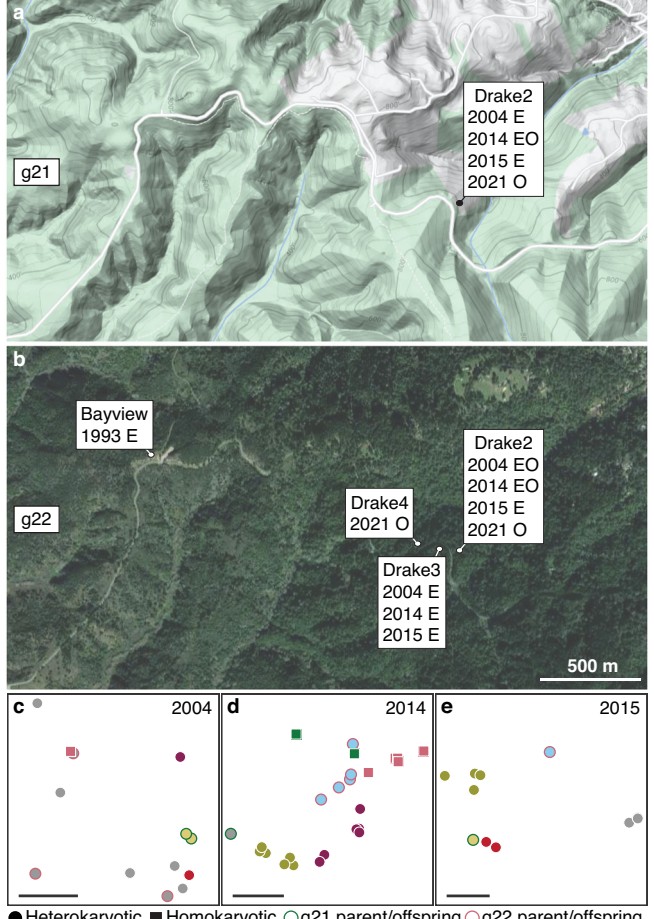

**Fig. 2 | Homokaryotic individuals at Point Reyes National Seashore, California, USA. a, b** Sites and years from which individuals g21 and g22 were collected. **a** Topography map and g21. **b** Satellite image and g22 (Google, ©2022). O: Site of homokaryotic mushrooms, E: Site of heterokaryotic mushrooms housing the g21 or g22 nucleus, either the parent or offspring of g21 or g22. **c–e** Fine-scale map of every mushroom collected from Drake2 between 2004–2015. Mushrooms are represented by circles (heterokaryotic mushrooms) or squares (homokaryotic mushrooms of g21 or g22) and shapes of the same color are genetically identical. Gray marks genetic individuals consisting of only a single mushroom. Heterokaryotic parent or offspring of g21 or g22 are outlined green or rose pink. Scale bars: 5 m.

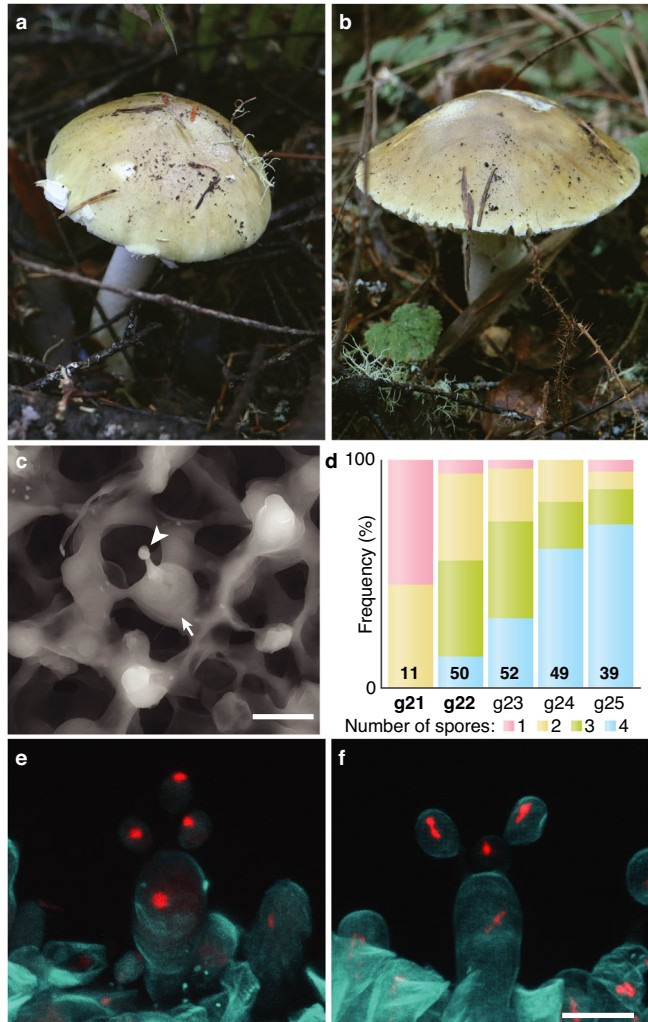

**Fig. 3 | Morphology of heterokaryotic and homokaryotic sporocarps. a** Heterokaryotic sporocarp found in 2021. **b** Homokaryotic sporocarp found in 2021. **c** Scanning electron microscopy of a unisporic basidium from a homokaryotic sporocarp. Arrowhead: immature spore; arrow: basidium. **d** Frequency spectra of the number of spores per basidium among five individuals. Homokaryotic individuals are bold (g21 and g22). Number of basidia counted indicated within each column. **e, f** Z-stack composite image of confocal microscopy of trisporic basidia from heterokaryotic (**e**) and homokaryotic (**f**) sporocarps; in (**f**) the basidium was more mature and nuclei are dividing. Red: Vybrant Orange (nuclei); cyan: Calco-fluor White (cell wall). Scale bars: 10 µm.

pheromone receptor locus (P/PR) and a homeodomain locus (HD). Successful sexual reproduction requires two fusing, haploid mycelia to carry different alleles at P/PR and HD, a system termed tetrapolar heterothallism. In bipolar heterothallic systems, either the P/PR and HD are linked, or one locus (usually P/PR) is no longer involved in mating[8].

We discovered *A. phalloides* possesses a bipolar mating system (Supplementary Discussion). As anticipated, we identified two putative mating type pheromone receptor (*PR*) genes (Fig. 4a, Supplementary Figs. 5a, 6, and Supplementary Note 3); however, we found that among *Amanita* species the number of putative *PR* genes is not consistent (ranging from two to five), and *PR* genes reside in a genomic region only weakly syntenic across the genus (Supplementary Fig. 5a). Unexpectedly, we were unable to identify genes predicted to encode pheromones (P genes) near the *PR* genes in *A. phalloides*. Instead, two apparent homologs of pheromones, each with the canonical -CaaX motif at its C-terminal, are located in other regions of the genome (Supplementary Fig. 7 and Supplementary Note 4). MAT loci are typically highly diverse, the result of frequency-dependent selection[19–21], but in *A. phalloides*, putative *PR* genes exhibit low genetic diversity

(Fig. 4a). Moreover, nine of the 25 heterokaryotic individuals tested carry identical copies of the *PR* genes. Finally, the two *PR* genes are orthologous to non-mating type-determining genes in other closely related species, as demonstrated in a species-tree-aware gene phylogeny (Supplementary Fig. 8 and Supplementary Note 5), although one of the *PR* genes is orthologous to a mating type-determining gene in a distant relative, *Cryptococcus neoformans* (Supplementary Fig. 9). The apparent absence of pheromones, as well as low genetic diversity, functional homozygosity of *PRs* in heterokaryotic individuals, and orthology between *PRs* and non-mating type-determining genes in other fungi, each suggests the *PR* genes are not involved in mating type determination. The irrelevance of *PR* genes to mating type determination is also observed in other basidiomycetes; in these fungi the pheromone receptor signaling pathway is hypothesized to be auto-activated by self-secreted pheromones[22, 23]. Autoactivated PRs from self-secreted pheromones, constitutively active PRs, as well as a bypass in the molecular pathway of sexual development are each alternative

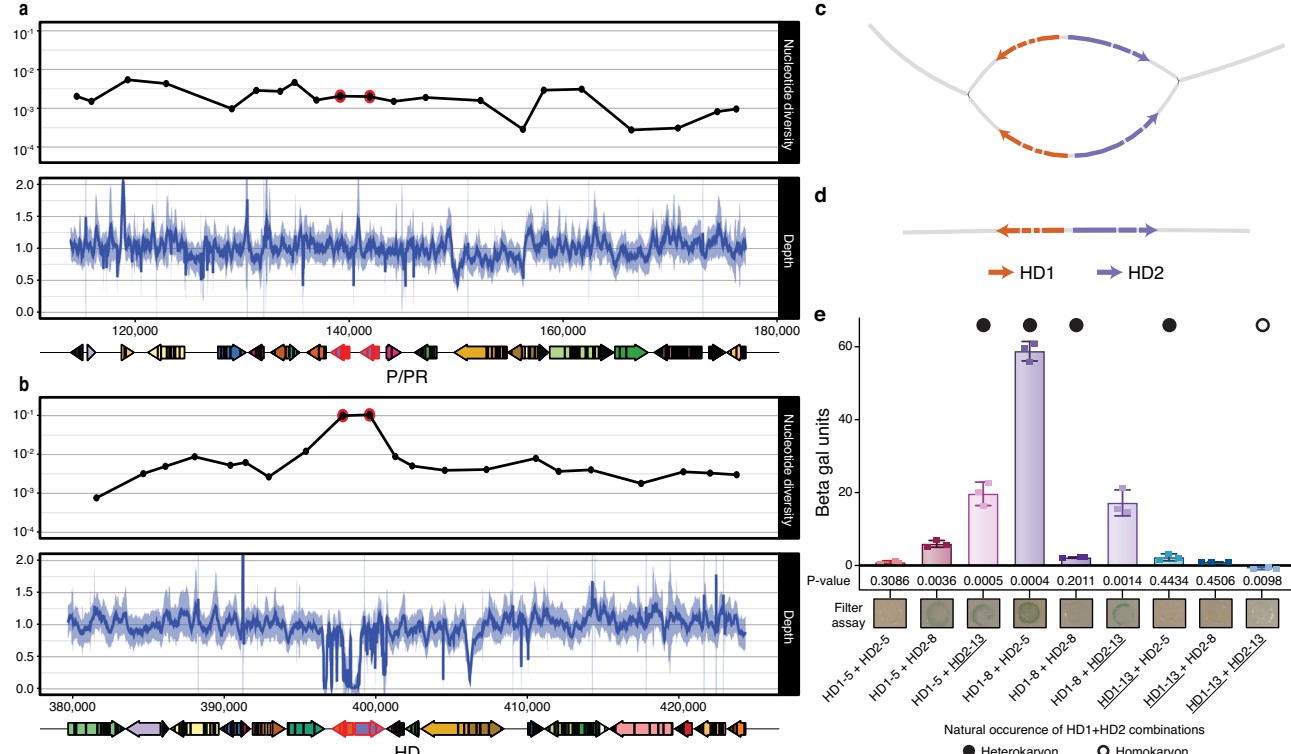

**Fig. 4 | Genomic features of putative mating loci and evidence for interactions between *HD* genes. a**, **b** Nucleotide diversity, sequencing depth and gene orientation of P/PR (**a**) and HD (**b**) loci. *PR* and *HD* genes outlined in red. Nucleotide diversity calculated from de novo genome assemblies; note diversity increase and depth drop at HD locus. **c**, **d** Assembly string graphs of the HD locus of heterokaryotic (**c**) and homokaryotic (**d**) sporocarps. Homokaryotic genomes house one copy of the locus. **e** Yeast two-hybrid β-galactosidase reporter activities from strains harboring HD1 and HD2 proteins of different alleles (designated 5, 8, and 13; HD1s fused with a Gal4 activation domain and HD2s fused with a Gal4 DNA binding domain; allele 13 (underlined) identified in a homokaryon (g22)). Closed circles: HD1 and HD2 combinations found in heterokaryotic mushrooms; Open circle: HD1 and HD2 combination found in homokaryotic mushroom. Error bars: standard errors of the means (each includes three biological replicates). P-values are determined by comparing *LacZ* expression in β gal units (Miller Units) to each respective BD plasmid co-transformed with the empty AD plasmid in an unpaired two-sided Student's t-test (d.f. = 4; n = 3).

hypotheses for the irrelevance of PRs in mating type determination in *A. phalloides*.

In contrast to the *PR* genes, *HD* genes in *A. phalloides* appear typical of MAT loci in agaricomycete fungi. In a typical MAT locus, there are two *HD* genes, designated as *HD1* and *HD2*. If the alleles of *HD1* and *HD2* in the nuclei of a heterokaryon encode compatible proteins, the proteins can form heterodimers, and heterodimers function as transcriptional regulators to promote mating and sexual development[24]. Across the genus *Amanita*, the HD locus also consists of two *HD* genes (Fig. 4b and Supplementary Fig. 5b). Within *A. phalloides*, the numbers of raw reads mapped onto the reference genome at the two *HD* genes were very low, compared to adjacent genes (Fig. 4b), suggesting HD alleles are highly diverged from each other. To extract the precise sequence of the alleles of each gene in each individual, we de novo assembled a genome for each Californian and Portuguese sporocarp (*N* = 77) (Fig. 4c, d, Supplementary Fig. 10, and Supplementary Notes 6 and 7). In all assembled genomes, we observed nucleotide diversity peaking at the *HD* genes, a hallmark of diversifying (frequency-dependent) selection (Fig. 4b). As expected, both the HD1 and HD2 proteins were predicted to possess canonical homeodomains (Supplementary Fig. 11), and only the *HD1* genes encode nuclear localizing signal (NLS) peptides. Models of predicted heterodimers suggest that HD1 and HD2 interact with one another via their N-termini (Supplementary Fig. 11 and Supplementary Note 3). The pattern demonstrates a congruence between *A. phalloides HD* genetic signatures and the known functionality of *HD* genes in *Coprinopsis cinerea*[24]; in *Co. cinerea* the NLS on HD1 is required to import HD2 into the nucleus where it can act. In the aggregate, our data document *A.*

*phalloides* as a bipolar species in which mating types are determined by a single MAT locus containing only *HD* genes.

Four mechanisms can explain the development of a sporocarp from a haploid fungus[9] (but see also Supplementary Discussion and ref. 25): (1) gene conversion from a silent mating type locus to create compatible mating loci, (2) gene duplication of the mating type locus followed by divergence to generate compatible mating loci within a single genome, (3) self-compatibility of genes within a mating type locus, or (4) drivers enabling a bypass of mating type control[9]. Neither gene conversion nor gene duplication can explain the homokaryotic *A. phalloides* sporocarps because we did not find multiple copies of either *HD* gene within any homokaryotic genome. To test for self-compatibility within the *A. phalloides* MAT locus, we carried out protein-protein interaction tests between the HD1 and HD2 proteins. We hypothesized that if the HD1 and HD2 proteins from the homokaryotic sporocarp are self-compatible, they would interact with one another in a yeast two-hybrid assay (Supplementary Tables 3 and 4). We evaluated HD1 and HD2 proteins from three different MAT alleles; two from heterokaryotic sporocarps (alleles designated 5 and 8), and one from a Californian homokaryotic sporocarp (allele 13). When experimenting with the HD1 and HD2 alleles from heterokaryotic sporocarps, we anticipated that proteins from at least one of the HD1-HD2 pairs drawn from the different MAT alleles would interact but that the HD1 and HD2 proteins from the same MAT allele would not produce any significant signal of an interaction. This was, indeed, the case and is congruent with findings in other fungal systems (Fig. 4e, Supplementary Fig. 12, and Supplementary Note 8)[24,26,27]. The HD1 and HD2 proteins from the homokaryotic sporocarp also did not produce any

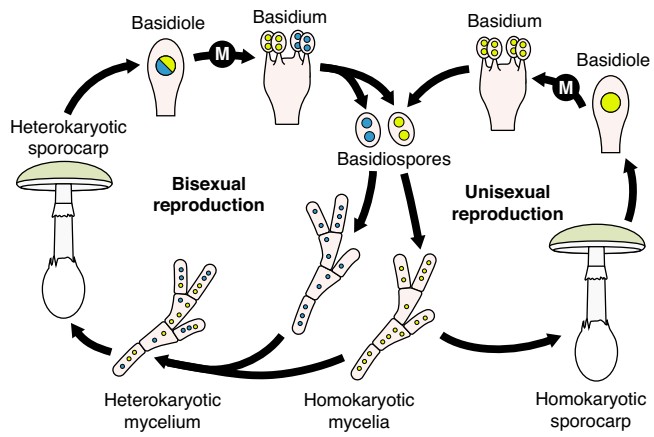

**Fig. 5 | Current model of the life cycle of *Amanita phalloides*, illustrating a bisexual reproductive cycle (left) and a unisexual reproductive cycle (right).** M marks meiosis. Following meiosis, the single nucleus of each spore undergoes a mitotic division and so each spore carries two (genetically identical) nuclei.

significant signal of an interaction with each other or with themselves under the conditions tested (Fig. 4e, Supplementary Fig. 13, and Supplementary Note 8). Thus, the heterodimerization of HD proteins from a homokaryotic MAT allele is unlikely to enable sporocarp development. Our finding suggests other mechanisms besides heterodimerization enable the development of homokaryotic mushrooms.

### Geography of unisexual individuals

To discover whether homokaryotic *A. phalloides* sporocarps can be found at other sites or in other ranges, we Sanger sequenced the variable beta-flanking gene adjacent to *HD1* as well as ten other mostly unlinked conserved genes from an additional 109 sporocarps collected from three sites in and around Berkeley, California ($N = 15$), from populations introduced to New Jersey ($N = 15$) and New York ($N = 10$), and from Canada, where introduced death caps grow on Victoria Island ($N = 8$). From its native range, Europe, we sequenced a population of death caps collected near Montpellier, France ($N = 12$), from two sites in Norway ($N = 6$), twelve sites in the UK ($N = 14$), four sites in Austria ($N = 13$), two sites in Estonia ($N = 2$), two sites in Hungary ($N = 12$), and one site in Switzerland ($N = 2$) (Supplementary Data 2). The sequence data allow us to evaluate whether collected sporocarps lack heterozygosity. With this approach, the probability of a heterokaryotic sporocarp being misidentified as homokaryotic is estimated as lower than 0.2%. We discovered no homokaryotic sporocarps at any other site (Supplementary Table 5).

Homokaryotic sporocarps appear to be extremely rare in nature, but in California homokaryotic individuals can span large territories, and sporocarps of g22 were found up to 200 m apart (Fig. 2b). Homokaryotic individuals have persisted for at least seven (2014–2021; g21) and up to 17 (2004–2021; g22) years. Moreover, an heterokaryotic herbarium specimen (individual g7) collected in 1993 at a site approximately 1.6 km away from our first g22 collection houses the g22 nucleus and is either the parent or offspring of individual g22. If g7 is an offspring, the g22 nucleus would have persisted for nearly 30 years in California and would have a much wider territory than we have discovered using only the presence of its homokaryotic mushrooms as a guide. Regardless of whether g7 is the parent or an offspring, the distances among the sites suggest unisexual sporocarps develop viable spores capable of dispersal; *Amanita* spp. grow slowly and otherwise the largest *A. phalloides* individual we have ever recorded at these same sites is less than nine meters across. Homokaryotic mycelia are considered an ephemeral stage of agaricomycete life cycles[28], and

our discovery of nuclei able to live alone (as homokaryons) as well as with other nuclei (in heterokaryotic mycelia) is surprising. Its apparent independence recalls nuclear dynamics in other fungi[29].

### Comparing *A. phalloides* to other basidiomycete fungi

Our discovery adds to a scarce literature documenting deviant life cycles among mushrooms growing in nature. Before the development of modern genetic tools, fungi were often sorted as either heterothallic (reproducing with another individual) or homothallic (reproducing without another individual)[30], and what we have discovered would have been named homothallism. However, the terminology used to describe sex in fungi is complex[31] and occasionally inconsistent. Categories including homothallism may be better understood as umbrella terms encompassing multiple kinds of more complex mating systems, including for example primary homothallism and pseudohomothallism[12]. More rarely observed phenomena are also a feature of homothallism, and two phenomena in particular may be relevant to *A. phalloides*. Monokaryotic fruiting is the development of sporocarps from a mycelium not possessing compatible mating types; basidiospores are generated mitotically (and each basidium bears two spores, each with one of the mitotic products)[15]. Unisexual reproduction is also the development of sporocarps from a mycelium not possessing compatible mating types, but basidiospores are generated meiotically (each basidium should bear four spores, each with one of the meiotic products)[10,32,33]. In the last decade, these apparently unusual reproductive mechanisms have been reported in multiple species including *Cyclocybe aegerita*, *Cy. parasitica*, and *Volvariella volvacea*[34–37]. The nuclear dynamics within the basidia of homokaryotic *A. phalloides* sporocarps are evidence of unisexual reproduction, despite the inconsistent numbers of basidiospores on basidia. We hypothesize single basidiospores occasionally germinate into haploid mycelia able to develop mushrooms and sporulate via endoduplication. Some of the offspring of these mushrooms mate while others do not, and the cycle repeats (Fig. 5).

### The spread of invasive mushrooms

The spread of *A. phalloides* in California is likely facilitated by its ability to sporulate without mating with another individual. The fungus is both unisexual and bisexual, revealing a previously unsuspected reproductive flexibility in a natural population of death caps. Its life cycle appears most similar to the life cycles of species of *Cryptococcus*, in which both unisexual and bisexual reproduction coexist. Unisexuality in both *A. phalloides* and *Cr. gattii* is associated with biological introductions[33], providing additional support for the selective advantage provided by self-reproduction in an introduced range, and revealing a profound similarity between plant, animal, and fungal invasions.

## Methods

### Mushroom collecting

Sporocarps were collected from various herbaria and during three expeditions to Point Reyes National Seashore (PRNS), California in 2004, 2014 and 2015, and in 2015 from three sites in Portugal. A total of 86 sporocarps were collected: 67 Californian sporocarps (one early herbarium sample dates to 1993), 11 Portuguese sporocarps, and eight sporocarps from other European countries (Supplementary Data 1). Specimens of sporocarps are deposited in the fungarium in Pringle laboratory. The Californian specimens collected from 2004 to 2015 were mapped.

### DNA extraction for genome sequencing

To extract DNA for genome sequencing, approximately 50 mg of cap tissue from each sporocarp was placed in a 2.0 ml microcentrifuge tube with four to five 3 mm glass beads and macerated using a MiniBeadbeater-8 (BioSpec Products Inc., Oklahoma) set at 75% speed

for 1 min. 700 μl of CTAB buffer (2% cetyltrimethyl ammonium bromide, 2% PVP, 100 mM Tris-HCl, 20 mM EDTA, and 1.4 M NaCl [pH8.0]) was added after maceration, and samples were left to incubate at 60 °C for one hour. Next, 700 μl of a 24:24:1, by volume, phenol:chloroform:isoamyl alcohol solution was added to each sample and samples were gently mixed at room temperature for 10 min, followed by centrifugation at room temperature at 13,000 rpm for 10 min. The aqueous phase (~650–700 μl) of each sample was then carefully transferred to a new 2.0 ml tube. 700 μl of the phenol:chloroform:isoamyl alcohol solution was again added to each sample, and samples were inverted and mixed at room temperature for 10 min, followed by centrifugation at room temperature at 13,000 rpm for 10 min, at which point the aqueous phase was again transferred to a new 2.0 ml tube. Approximately 1.4 ml of 100% ethanol was added to each sample, and samples were incubated at −20 °C for 45 min, and then centrifuged at 4 °C at 13,000 rpm for 10 min. The supernatant was discarded, and the pellet dried in a Savant DNA 120 SpeedVac Concentrator (Thermo Fisher, Massachusetts) at room temperature for 10 min, or until dry, and finally resuspended in 400 μl of 10 mM Tris-HCl (pH 8.0) and transferred to a new 1.5 ml tube. To further purify genomic DNA, 12 μl of RNase A (Qiagen, Germany) was added to each sample, and each sample incubated at 37 °C for an hour. 16 μl of 5 M NaCl and 860 μl of 100% ethanol were then added to each tube and the solution left to precipitate at −20 °C for one hour, after which each tube was centrifuged at 4 °C at 13,000 rpm for 15 min and the supernatant was discarded. A final washing was performed with 500 μl of 75% ethanol; solutions were centrifuged at 4 °C at 13,000 rpm for 10 min and the supernatant discarded. Finally, the resulting pellet was resuspended in 200 μl of 10 mM Tris-HCl (pH 8.0). 5 ml of an Oxygen AxyPrep Mag PCR Clean-Up kit (Fisher Scientific, Pennsylvania) was used per manufacturer instructions to remove any remaining impurities. DNA was stored at −80 °C until it was provided to the University of Wisconsin-Madison Biotechnology Center.

### Genome library construction, sequencing, and read filtering

DNA was sent to the University of Wisconsin-Madison Biotechnology Center. Its concentration was checked using the Qubit® dsDNA HS Assay Kit (Life Technologies, Grand Island, NY). Samples were prepared according to the TruSeq Nano DNA LT Library Prep Kit (Illumina Inc., San Diego, California, USA) with minor modifications. Samples were sheared using a Covaris M220 Ultrasonicator (Covaris Inc, Woburn, MA, USA) and were size selected for an average insert size of 550 bp using SPRI bead-based size exclusion. The quality and quantity of the finished libraries were assessed using an Agilent DNA1000 chip and Qubit® dsDNA HS Assay Kit, respectively. Libraries were standardized to 2 nM. Cluster generation was performed using the Illumina Rapid PE Cluster Kits v2 and the Illumina cBot. Paired-end, 251 bp sequencing was performed, using Rapid v2 SBS chemistry on an Illumina HiSeq2500 sequencer. Images were analyzed using the Illumina Pipeline, version 1.8.2.

Mean sequencing depth of each sample ranged from 10.56 to 150.86 (Supplementary Data 1; low depths characterized older specimens). Sequence data were filtered using Trim Galore! (ver. 0.4.5) (https://github.com/FelixKrueger/TrimGalore). Adapter trimming was set to the highest stringency so that even a single nucleotide of overlap with the adapter sequence was trimmed from a given read. After trimming, reads reduced to 100 bp or shorter and those with quality scores less than 30 were discarded.

To facilitate the assembly of a high-quality reference genome, a sporocarp from Coimbra, Portugal (10511) was also sequenced with a long-read technology. A Pacific Biosciences HiFi library was prepared using the Template Prep Kit 1.0 (Pacific Biosciences). Modifications included shearing with Covaris gTUBEs and size selecting with Sage Sciences BluePippin. The library was quantified using the Qubit™ dsDNA High Sensitivity kit. The library was sequenced on four

SMRTcells on a PacBio RS II Sequel platform, also at the University of Wisconsin-Madison Biotechnology Center DNA Sequencing Facility. PacBio sequencing resulted in a raw coverage of 47x with an N50 read length of 6,310 bp.

### Reference genome assembly

After testing five different genome assembly pipelines, we used an in-house hybrid approach to assemble the final reference genome. First, Illumina data of 10511 were subjected to a second round of filtering using Trimmomatic ver. 0.35[38] with the following parameters: ILLUMINACLIP:TruSeq3-PE-2.fa:2:30:10 CROP:245 LEADING:30 TRAILING:30 SLIDINGWINDOW:4:25 MINLEN:100. PacBio data were filtered to remove sequences shorter than 500 bp and error corrected with the Illumina data using FMLRC[39] and default settings. Error-corrected PacBio reads were then used to simulate 20x coverage 3 kbp insert size libraries with wgsim (https://github.com/lh3/wgsim) and parameter setting as follows: -e 0.0 -d 3000 -s 500 −1 100 −2 100 -r 0.0 -R 0.0 -S 123 -N 5000000. Illumina data and simulated long-range libraries were then assembled using AllpathsLG ver. 52400[40], setting HAPLOIDIFY=True. The resulting assemblies were subjected to further scaffolding using error-corrected PacBio data with the software LINKS v.1.8.5[41] with -d 2000,5000,10000,15000,20000 -t 20,20,5 and a k-mer value of 29. Scaffolds were extended and gap-filled using PBJelly v.15.8.24 and finally polished using Pilon v.1.2[42]. Polished assemblies were evaluated with QUAST[43] and BUSCO ver. 2 with the Basidiomycota database ver. 9[44]. The final assembly is 35.5 Mb, consisting of 605 scaffolds. The N50 and NG50 is 320 kbp and 184 kbp, respectively. The assembly encodes 1260 single-copy BUSCOs (94.4%).

### Variant calling and SNP filtering

After the reference genome was assembled, single nucleotide polymorphisms (SNPs) and insertions and deletions (indels) in all genomes were identified using the Genome Analysis ToolKit (GATK) software v3.8-0-ge9d806836[45], following GATK best practices. Illumina reads from each of the 86 Illumina genome libraries were first mapped to the hybrid reference assembly using BWA-0.7.9[46] with the following parameters: mem -M -t 8 -v 2. Mapping rates for *A. phalloides* specimens ranged from 20.0% to 95.3%, with a median mapping percentage of 86.1% (only older specimens mapped at less than 50%). The mapping rate of 10511's Illumina reads to the 10511 hybrid assembly was 93.8%. Duplicate reads were marked, and the GATK program Haplotypecaller was used to call variants simultaneously on all samples, using MODE = DISCOVERY and Type=GVCF. Because of the lack of known variants in *A. phalloides*, the raw variants were hard-filtered according to GATK's default parameters of the VariantFiltration program. The pipeline resulted in Variant Call Files (VCFs) containing 212,119 indels and 1,580,133 SNPs. To identify genetic individuals, we only used SNPs and refer to this VCF file as the "raw VCF".

To eliminate any SNPs called as the result of sequencing errors, the raw VCF was additionally filtered using VCFtools ver. 0.1.14[47]. All transposable elements, multi-allelic sites, and contigs with putative contamination were removed. First, transposable elements were identified using REPET ver. 2.5 and removed. Multi-allelic sites were both identified and removed, and putatively contaminated contigs were identified with CAT ver. 5.0.3 (https://github.com/dutilh/CAT; contigs 330, 313 and 581) and also removed. In addition, sites with a sequencing depth below a minimum depth of 85% of the per specimen mean depth, and above a maximum depth of $DP + 4\sqrt{DP}$ of the per specimen mean depth, were removed[48]. Next, variants were re-called based on allelic depth: if the ratio of allelic depths for a given site was between 0.25 and 0.75, that site was called as heterozygous (0/1), and ratios below and above were called homozygous reference (0/0) and homozygous alternate (1/1), respectively. We refer to the new VCF file as the "filtered VCF".

## Transcriptome sequencing

To sequence the transcriptome of *A. phalloides* for genome annotation, total RNA from sample 10721 was extracted using an RNeasy Mini Kit (Qiagen, Germany). We first macerated around 100 mg of tissue in 700 μl RLT Buffer and 7 μl β-mercaptoethanol with three 2.7 mm glass beads using a MiniBeadbeater-8 (BioSpec Products Inc., Oklahoma) for 1 min at 3000 rpm. We chilled the sample on ice for 1 min and macerated again for 1 min using the same settings. After centrifugation for 3 min at 14,000 rpm, 650 μl of supernatant was transferred to a new microcentrifuge tube. Then we added 650 μl of 100% ethanol and gently shook the tubes ten times to mix. We passed 650 μl of the solution through an RNeasy mini column with 14,000 rpm centrifugation for 15 sec twice. To clean up the RNA extract, the column was washed with 350 μL RW1 buffer with 14000 rpm centrifugation for 15 sec, and DNA was removed by incubating the column in 80 μL DNase solution (7 RDD buffer (Qiagen, Germany):1 DNase stock solution (Qiagen, Germany)) for 15 min, then washed with 350 μL RW1 buffer once and 500 μL RPE buffer twice with an additional 1.5 min centrifugation for the last wash. Finally, RNA was eluted with 30 μl RNase-free ddH$_2$O by incubating for 2 min and 14,000 rpm centrifugation for 2 min twice. We collected the flow-through solution and stored it at −80 °C prior to sequencing at the University of Wisconsin-Madison Biotechnology Center.

The purity and integrity of the total RNA was assessed via the NanoDrop One Spectrophotometer (ThermoFisher, Inc., Carlsbad, CA, USA) and Agilent 2100 Bioanalyzer (Agilent Technologies, Inc., Santa Clara, CA, USA), respectively. Stranded RNA libraries were prepared from samples that met the TruSeq™ Stranded Total RNA With Illumina® Ribo-Zero™ Plus rRNA Depletion input guidelines using the Illumina® TruSeq Stranded Total RNA with Ribo-Zero Plant kit (Illumina Inc., San Diego, California, USA). For each library preparation, cytoplasmic, mitochondrial and chloroplast ribosomal RNA was removed using biotinylated target-specific oligos combined with Ribo-Zero rRNA removal beads. Following purification, the RNA was fragmented using divalent cations under elevated temperature. Fragmented RNA was copied into first stranded cDNA using SuperScript II Reverse Transcriptase (Invitrogen, Carlsbad, California, USA) and random primers. Second strand cDNA was synthesized using a modified dNTP mix (dTTP replaced with dUTP), DNA Polymerase I, and RNase H. (The incorporation of dUTP quenches the second strand during amplification.) Double-stranded cDNA was cleaned up with AMPure XP Beads (1X) (Agencourt, Beckman Coulter). The cDNA products were incubated with Klenow DNA Polymerase to add a single 'A' nucleotide to the 3′ end of the blunt DNA fragments. Unique dual indexes (UDI) were ligated to the DNA fragments and cleaned up with two rounds of AMPure XP beads (0.8X). Adapter ligated DNA was amplified by PCR for 10 cycles and cleaned up with AMPure XP beads (0.8X). Final libraries were assessed for size and quantity using an Agilent DNA1000 Screentape and Qubit® 1X dsDNA HS Assay Kit (Invitrogen, Carlsbad, California, USA), respectively. Libraries were standardized to 2 nM. Paired-end 2x150bp sequencing was performed, using standard SBS chemistry (v3) on an Illumina NovaSeq6000 sequencer. Images were analyzed using the standard Illumina Pipeline, version 1.8.2.

We received 170,345,047 paired-end raw reads of 126 bp sequences. The raw data were trimmed using Trimmomatic with tags "ILLUMINACLIP:TruSeq3-PE-2.fa:2:30:10", "MAXINFO:30:0.4", and "MINLEN:75" to remove adapters, short reads, and unpaired reads. 142,428,127 read pairs were retained. As a preliminary assessment of data quality, we aligned the trimmed reads to the reference nuclear genome using HISAT2 ver. 2.1.0[49] setting a minimum intron length of 20 and a maximum intron length of 500. We observed an alignment rate of 81.5%.

## Reference genome annotation

We annotated the reference genome with GenSAS server[50] using the transcriptome as evidence to train several gene predictors and using EvidenceModeler ver. 1.1.1[51] to subsequently weigh and combine the predictions. We first de novo assembled the transcriptome using Trinity ver. 2.2.0[52] with --jaccard-clip flag. In addition, we performed a second assembly of the transcriptome, guided by the genome, by mapping raw reads to the genome while limiting the maximal intron length at 300, using HISAT2 and assembling the mapped reads using Trinity with --jaccard-clip flag. Before undertaking the next steps of our annotation pipeline, we identified repetitive regions in the genome with RepeatMasker ver. 4.0.7[53] on GenSAS, using rmblast, quick speed and a fungal repeat database. We also used RepeatModeler ver. 1.0.11[54] on GenSAS to identify novel repetitive regions. The repetitive regions were then masked from the genome based on the outputs from both RepeatMasker and RepeatModeler. We then generated gene models from the combined de novo and genome-guided transcriptome assemblies with PASA[55] using default settings on GenSAS. We also generated a refined set of gene models by using the *exonerate* and *prepare_golden_genes_for_predictors.pl* tools from the JAMg[56] pipeline to pick the best gene models from the original set.

To de novo predict nuclear genes, we used five gene predictors: GeneMark-ES ver. 4.38[57], BRAKER ver. 2.1.0[58], SNAP[59], AUGUSTUS ver. 3.3.1[60] and CodingQuarry ver. 2.0[61]. We ran GeneMark-ES on GenSAS with default settings. We also ran BRAKER on GenSAS but trained it with the mapped transcriptome generated from HISAT2 on GenSAS. We trained SNAP and AUGUSTUS with the refined set of gene, and trained CodingQuarry with the original set.

To combine gene predictions, we tuned the weights of the five different predictors for EvidenceModeler based on: their individual coverage of the benchmarking universal single-copy orthologs (BUSCOs) of basidiomycetes (OrthoDB ver. 9)[44], gene boundaries, fragmentation/fusion and intron lengths of each prediction result and the different runs of EvidenceModeler. After exploring different weight combinations, we gave AUGUSTUS, BRAKER, GeneMark-ES, CodingQuarry, SNAP, and PASA (transcript) weights of 5, 6, 6, 2, 5, and 10, respectively. Finally, we refined the gene models with PASA and used InterProScan ver. 5.29-68.0[62] and Pfam ver. 1.6[63] for functional annotation.

We identified 8746 gene models with our pipeline, including 95.0% of basidiomycetes' BUSCOs (slightly different from the 94.4% resulting from our assembly of the reference genome). Our number of gene models was lower than a previous annotation from Pulman et al.[64] (10,221 models), but our percentage of annotated BUSCOs was higher (in Pulman et al.[64], 92% of BUSCOs were annotated). The difference between our annotation and Pulman et al.'s[64] annotation may relate to assembly coverage. Our reference assembly was 35.5 Mb with an estimated 45.5 Mb genome (78% complete), whereas Pulman et al.'s[64] assembly was 40 Mb (without an estimated genome size). Our annotation pipeline may also be more conservative than Pulman et al.'s[64] MAKER pipeline; when we used the BRAKER gene predictor (a combination of AUGUSTUS and MAKER) in our annotation pipeline, it identified 9417 gene models. Finally, Pulman et al.'s[64] genome was sequenced from California while our reference genome was collected in Portugal. The difference in gene model numbers may also reflect some degree of biological diversity. Nevertheless, estimates of BUSCO coverage suggest our genome annotation performs better on conserved genes, ideal for identification of conserved mating type loci.

## Clone-correction

To understand which sequenced sporocarps were collected from a single genetic individual (from the same mycelium), we adapted methods from the R package *poppr*. Euclidean genetic distances were calculated between all pairs of sporocarps and visualized as a histogram (Supplementary Fig. 1a). A distinct peak in the histogram, near zero and apart from a second distinct peak, marks the sporocarps belonging to single individuals[65]. The 86 mushrooms resolved into 37 individuals, including 27 individuals consisting of only one mushroom

and ten individuals consisting of multiple mushrooms (Supplementary Fig. 1a and Supplementary Data 1). No heterokaryotic individuals spanned different collecting sites. We also used a second approach developed by us to identify kinship and used it to confirm the results generated using Euclidean genetic distances (see section "Kinship analysis"); both approaches identified the same genetic individuals.

## Heterozygosity estimation, k-mer analysis and allele frequencies

We tested whether any sporocarps without heterozygosity are found among our sequenced individuals by comparing individuals' heterozygosities, k-mer distributions and allele frequency graphs. Because a single mycelium can generate multiple mushrooms, we first sorted the sporocarps into genetic individuals and then we counted the homozygous sites for each sporocarp using the filtered variant calling format (VCF) file and VCFtools ver. 0.1.16[47]. The heterozygosity of each sporocarp was next estimated with

$$het_i = \frac{\frac{N_{sites(i)} - N_{homozygous(i)}}{N_{sites(i)}}}{\frac{N_{genome} - N_{repetitive}}{N_{VCF}}} \qquad (1)$$

where $N_{sites(i)}$ was the number of non-missing sites in sample $i$. $N_{homozygous(i)}$ was the number of homozygous sites in sample $i$. $N_{genome}$, $N_{repetitive}$ and $N_{VCF}$ were the assembly size, the size of the masked repetitive regions identified by REPET, and the number of sites in the VCF file. The heterozygosity of each genetic individual was calculated as the mean of the heterozygosities of all sporocarps making up that individual.

To further explore the zygosities of individuals with low estimated heterozygosities, we first used k-mer analyses. Because herbarium sporocarps (most non-Portuguese European sporocarps and the 1993 Californian sporocarp) had lower DNA quality than the newer specimens collected by us, we only compared our samples from California or Portugal collected after 2000. For k-mer analysis, we used BBMap ver. 38.73[66] to generate k-mers of 23 bp and count the number of each different k-mer. The k-mer frequency distribution of each genome was normalized by its largest peak.

As a second approach to probe zygosities, we investigated per individual allele frequency. We plotted histograms of the sequencing frequencies of alleles of putatively heterozygotic sites from the filtered VCF. Heterozygotic diploid individuals should display a peak at 0.5 because the two alleles at a site normally have a similar sequencing depth. To be sure sequencing depth was unaffected by the variant re-calling step, we also reran this analysis using the same filters used to generate the filtered VCF but without variant re-calling.

## Kinship analysis

To test if the homokaryotic individuals discovered in California are mating with other individuals in the population, we estimated kinships between homokaryotic individuals and heterokaryotic individuals with KIMGENS, a population-structure-robust estimator[16]. To generate a set of SNPs for KIMGENS, we first removed sites called as heterozygous in any homokaryotic sporocarp from the filtered VCF file. Following KIMGENS, we set a threshold at 0.75 to determine which mushrooms belong to the same homokaryotic individual(s) and at 0.4375 to determine which mushrooms belong to the same heterokaryotic individual(s). The results from KIMGENS and Euclidean distances were identical.

In subsequent analyses we generated a consensus genotype for each genetic individual consisting of multiple mushrooms. We next compared kinships among the 37 genetic individuals. Because kinship is defined as the probability of IBD between two randomly chosen alleles, one from each individual in a pair of interest, the immediate kins of a homokaryotic individual (i.e. its parent and heterokaryotic offsprings) will share kinships of 0.5 with the homokaryotic individual.

## Population-level phylogeny reconstruction

Although kinship estimates suggest homokaryotic individuals are mating, they cannot exclude the possibility of homokaryotic individuals belonging to other (as yet unidentified and unnamed) reproductively isolated groups (e.g., a cryptic species). To explore this hypothesis, we used a coalescent-based phylogeny reconstruction method. We called BUSCOs of Agaricales from the reference assemblies of *A. phalloides* and *A. subjunquillea* (the closest relative of *A. phalloides*) with BUSCO ver. 5.2.2 using *Laccaria bicolor* for the AUGUSTUS species parameter. Then, the DNA sequences of the common single-copy BUSCOs of each individual were called from filtered VCF files with *vcf-consensus* from VCFtools with IUPAC codes. All sequences for a BUSCO were aligned with MAFFT ver. 7.427[67], and a phylogeny was reconstructed using IQ-TREE ver. 2.0.6[68] with a nuclear substitution model identified by ModelFinder Plus[69] and bootstrapped 1000 times with a hill-climbing nearest neighbor interchange (NNI) search[70]. Then, we controlled for highly related individuals in the phylogenies by using a Markov clustering analysis (MCL). To avoid excess pruning, any pairwise kinship lower than 0.1 was recoded to 0 and the inflation rate of MCL was set to 1.5. To ensure one homokaryotic individual was included in each of the final phylogenies, one individual was picked for each cluster manually. Since the two homokaryotic individuals were related, this pruning process retained only one homokaryotic individual: g22. We summarized the phylogenies of BUSCO genes using the "constrained-search" branch of ASTRAL ver. 5.6.9[71,72]. We reconstructed two phylogenies: one unconstrained, and the other constrained by forcing *A. subjunquillea* and the homokaryotic individual to form a single cluster.

## Identification of mating type loci across the genus *Amanita*

To establish if the two homokaryotic individuals possess a single mating type allele, we first identified mating type loci in the reference genome and across the genus *Amanita*. Basidiomycetes are typically tetrapolar and so we used the protein sequences of the mating type homeodomain proteins (HD; XP_001829154.1, XP_001829153.1) and mating type pheromone receptors (PR; AAQ96344.1, AAQ96345.1) of *Coprinopsis cinerea* as queries for BLASTp[73], searching within the predicted proteome of *A. phalloides* (JAENRT000000000.1) to identify homologs. To identify syntenic regions, we searched for the *HD* and *PR* genes, as well as homologs of ten genes located up- and downstream of the two loci, searching in the predicted proteomes of *A. brunnescens* (JNHV00000000.2), *A. polypyramis* (JNHY00000000.2), *A. muscaria* var. *guessowii* (JMDV00000000.1) and *A. inopinata* (JNHW00000000.2). After completing structural analyses of proteins (see below), we manually reannotated the genes of the two loci.

To further explore the biology of putative pheromone receptor genes, we inferred their orthology using a species-tree-aware phylogeny reconstruction method (see section "Interspecies phylogeny of *PR* genes"). We based our phylogeny on the GO terms of the pheromone receptors of *A. phalloides* confirmed to be "GPCR fungal pheromone mating factor, STE3"; these were the only genes annotated as such.

## Identification of pheromones in *A. phalloides*

The genes of pheromones (*P* genes) are shorter than canonical genes and our annotation pipeline was unable to identify pheromones within the pheromone and pheromone receptor (P/PR) locus. To identify pheromones, we first used the ORFfinder[74] in NCBI to predict open reading frames (ORFs) longer than 30 bp between 5 kbp upstream and downstream of the *PR* genes. Then we looked for any ORFs with ER/DR (N-terminal cleavage) and -CaaX (farnesylation) motifs. However, no pheromones close to the *PR* genes were identified, and so we used protein sequences of the pheromones of *A. muscaria* (which appears to have retained its *P* genes at the P/PR locus) taken from the JGI genome portal (gene ID: 163418 and 163420) to search in the *A.*

*phalloides* genome assembly with tBLASTn. These pheromone gene predictions were further refined by comparing the predictions against transcripts.

### Sequencing depth of mating type loci

Fungal mating type locus alleles are in general highly polymorphic, and read mapping algorithms often perform poorly on them, resulting in erroneous variant calls. To understand if poor mapping is a major concern for the mating type loci of *A. phalloides*, we used SAMtools ver. 1.5[75] to estimate the sequencing depth of each locus and normalized each locus's average depth to one.

### Identification of homeodomains proteins in *A. phalloides* I: De novo assembly for HD calling among all samples

We discovered the *A. phalloides* HD locus does in fact map poorly, and so we chose to de novo assemble the genomes of each sporocarp as the first step towards identifying the HD locus of every genetic individual. In a graph-based genome assembly pipeline, different alleles of a locus in a heterokaryotic genome will be assembled into two unitigs (equivalent to a contig without conflicting nucleotides). Because these two unitigs represent two alleles, they are also called haplotigs. In a successful assembly, the two haplotigs each connect with the rest of the genome on either side, and so form a "bubble" within the assembled genome. We took advantage of assembly bubbles to identify the *HD* genes.

We first trimmed the raw reads with Trimmomatic ver. 0.35[38] to remove the adapters tagged as "ILLUMINACLIP:TruSeq3-PE-2.fa:2:30:10" and "MINLEN:75". We then assembled each sample with the de Bruijn graph-based assembler SPADES ver. 3.13.1[76], using both paired-end reads and unpaired reads (k-mer size=21). To identify the unitigs containing *HD* genes (the HD haplotigs), we used the DNA sequences of *HD* genes from the reference genome of *A. phalloides* as queries to search within the assembly graph with Bandage ver. 0.8.1[77]. To confirm the HD haplotigs as the two alleles of a sample, we also identified the connected unitigs five unitigs away/around the HD haplotigs on either side of them. We grouped sporocarps into one of five categories (Supplementary Fig. 10): 1. closed bubble: the HD haplotigs attached to the same unitigs on both sides; 2. open bubble: the HD haplotigs only attached to the same unitig on one side; 3. detached: the HD haplotigs were not linked to each other; 4. complexed: more than one unitig on either side of the HD haplotigs were linked back to the haplotigs; and 5. odd-number unitig: when only one or more than two unitigs were identified as HD haplotigs. For individuals appearing to have only one HD haplotig, we evaluated assembly quality by mapping the raw reads back to the HD haplotig using the BWA MEM algorithm.

### Identification of homeodomains proteins in *A. phalloides* II: Annotation of *HD* genes in each genome

Using the genome assemblies of sporocarps, we annotated the *HD* genes of every specimen collected between 2004 and 2015 from California and Portugal. We first used four different strategies to predict the genes within each genome's HD haplotigs: a pretrained AUGUSTUS, a self-trained AUGUSTUS, SNAP and CodingQuarry. For pretrained AUGUSTUS, we used the AUGUSTUS web interface[78] to predict genes based on a *Laccaria bicolor* gene model. For self-trained AUGUSTUS, we used the gene models chosen from whole genome annotation and trained AUGUSTUS ver. 3.3.3 on Galaxy server[79]. For SNAP and CodingQuarry, we followed the strategies used in whole genome annotation described above.

Gene predictors sometimes produce faulty annotations, so next we manually reannotated the introns, and start and stop codons, of the *HD* genes of 10721 (the same specimen of which transcriptome was sequenced) based on its mapped transcriptomic reads. Fungi normally use the first start codon in a transcript as the translation initiation site[80], and therefore we annotated the first in-frame AUG of the majority of transcripts as the start codon. We noted that 15 of 986 transcriptomic reads extend toward the upstream of most other reads of the *HD2* gene. These 15 reads include another putative start codon starting at the third nucleotide. But because of the low depth of these reads and the close position of the alternative putative start codon to the 5′ end of the transcript, we did not consider these 15 reads as encoding the true start codon of *HD2*. Using this information, we later manually reannotated the *HD* genes for each sporocarp's genome by aligning and comparing among protein sequences. Because the sporocarps making up a genetic individual did not always have the exact same DNA sequences, we first aligned the protein sequences with MAFFT and reverse translated the alignment to coding DNA sequences (CDSs). Then we generated the consensus CDS for each allele of each individual.

To compare sporocarps to each other, and to discover whether HD alleles are shared across different genetic individuals, we reconstructed phylogenies for the *HD1* and *HD2* alleles. We translated the already aligned CDSs back to proteins and built a phylogeny for each of the two *HD* genes with RAxML-HPC ver. 8.2.9 with the best protein substitution models determined by ModelTest-NG[81] and bootstrapped for 100 times.

### Structural analyses of pheromone receptor and homeodomain proteins

Next, we sought to understand if the *PR* and *HD* genes encode canonical mating type determining proteins by analyzing protein structures. We first explored whether either of the *PR* genes encodes the canonical structures of the seven-transmembrane domain in G-protein coupled receptors (GPCRs) or signal peptides. We identified the sequences from the reference genome annotated as *PR* genes. To identify transmembrane helices, we submitted sequences to the CCTOP server[82] with TM filter. To identify signal peptides, we used SignalP 5.0[83] and searched for eukaryotic signal peptides. Then we predicted the general structures of the *PR* genes using Alphafold2[84] on Cosmic[2,85] and its full protein database (February, 2022).

Then, we chose to predict the HD protein structures of heterokaryotic g19 (alleles 8 and 13), g33 (alleles 5 and 8) and homokaryotic g22 (allele 13), and targeted nuclear localization signal (NLS) peptides and homeodomains. To identify the NLS, we used cNLS Mapper[86] with a cut-off score of 5.0, and we only searched for bi-partite NLS within 60 amino acids of either terminal of the protein. To identify homeodomains, we used RaptorX-Property[87] to first identify alpha helices using a 3-class classification (of alpha-helix, beta-sheet or coil). Then we searched for alpha helices homologous to the homeodomains. To explore potential interactions between HD proteins, we used Alphafold-Multimer[88] on Cosmic[2] with its full protein database to predict the heterdimeric structure of HD1-8 and HD2-5 which were encoded by genes on the different alleles of a single individual: g33 (February, 2022). The protein structure was visualized with PyMol ver. 2.4.1[89] and the predicted alignment errors (PAEs) were extracted with paem2png.py (https://github.com/CYP152N1/plddt2csv).

### Nucleotide diversity of putative MAT loci

To test if the *PR* and *HD* genes are under diversifying selection, we compared their nucleotide diversities to the diversities of upstream and downstream genes. Because the *HD* genes were identified using a different approach from other genes, we were unable to filter SNP sites using a single protocol. Therefore, we used the unfiltered VCF of only heterokaryotic sporocarps from California or Portugal, and only genomes for which the HD locus was assembled correctly. We used the R package *pegas*[90] to calculate the nucleotide diversities of the ORFs of *HD* genes. We used VCFtools with the --site-pi flag to calculate the nucleotide diversities of other genes and correct for gene length as necessary. In this analysis, we did not clone correct.

## Interspecies phylogeny of *PR* genes

Because no substantial diversity was present within the *PR* genes of our population genomics dataset and many fungal species have *PRs* that do not determine mating type, we investigated the orthology of *A. phalloides* and other model systems' PR proteins using a species-tree-aware gene phylogeny. We first harvested the PR sequences of species listed in Coelho et al. (2017)[8] from the JGI genome portal[91] using the gene ontology term for mating type factor pheromone receptor activity (GO: 0004932), and sequences of two ascomycete pheromone receptors (*Saccharomyces cerevisiae* and *Pneumocystis carinii*) from NCBI (Acc. No. P06783.1 and AAG38536.1). We then used MAFFT to align the protein sequences of the retrieved pheromone receptors and our five *Amanita* species, trimming the alignment with trimAl ver. 1.4.rev15[92]. Finally, we reconstructed a species-tree-aware gene phylogeny with GeneRax ver. 2.0.4[93] using the best substitution model identified by ModelTest-NG, the species tree as described in Coelho et al. [8], and an undated duplication-loss model.

## Imaging

To compare the number of spores per basidium in homokaryotic and heterokaryotic *A. phalloides*, we imaged the basidia of five individuals (g21–g25, which are specimens of high quality) using a scanning electron microscope (SEM). We cut lyophilized gill tissue into pieces approximately $5 \times 5$ mm$^2$ in size and mounted tissue directly to a sample stub with conducting tape. We observed basidia under a FEI Quanta 200 SEM using the environmental SEM settings with a 25 kV electron beam with a spot size of 4.5 under 2.5 Torr, 5 °C, ×5000 magnification, and approximately 10 mm of working distance.

To understand the organization of nuclei within basidia, we first rehydrated and fixed lyophilized gill tissues in 10% neutrally buffered formalin for an hour at room temperature, followed by a single wash in ddH$_2$O. Then we stained tissues with 2 µg/ml Calcofluor White (Sigma-Aldrich, Missouri) in darkness for 20 min and subsequently washed the tissues with water. We hand-sliced gill thinly with a razor blade on a glass slide and stained them with a 5X Vybrant Orange dye (Thermo Fisher, Massachusetts) in darkness for 30 min under a coverslip.

We also sought to understand the organization of nuclei within homokaryotic and heterokaryotic mycelia, and so we cut the pith of fresh stipe tissues from samples collected in 2021 and fixed the tissues as described above, later splitting the mycelia with forceps and scalpels. We stained the tissues with a freshly mixed dye containing 2 µg/mL Calcofluor White and 4X Vybrant Orange for 30 min in darkness under a coverslip.

To visualize nuclei of basidia and mycelia, slides were mounted under an LSM 710 confocal microscope (Zeiss, Germany) using a C-Apochromat 40X/1.20 W Korr objective lens. We used two channels to separate the fluorescent signals from Calcofluor White and Vybrant Orange. For Calcofluor White, we used a wavelength of 405 nm for excitation and detected the emission from 394 to 515 nm. For Vybrant Orange, we used a wavelength of 514 nm for excitation and detected the emission from 525 to 678 nm. The pinhole size was set to one airy unit for the longer wavelength. Because nuclei often cannot be captured on a single optical section, we performed z-stacking with a step size of 0.49 µm and constructed 2D images with maximum intensity projection using the Z project function in FIJI[94]. Slices were removed when bacterial contaminants overlap with hyphal compartments. Images were lastly cleaned up with denoise function in FIJI[94].

We successfully imaged three homokaryotic trisporic basidia, five heterokaryotic trisporic basidia, four heterokaryotic tetrasporic basidia, seven homokaryotic hyphae and four heterokaryotic hyphae.

## Yeast two-hybrid of *HD* genes

We conducted a yeast two-hybrid experiment to test whether proteins encoded by the *HD1* and *HD2* of different alleles in a heterokaryotic individual, and from the same allele in a homokaryotic individual, can form a heterodimer. For the experiment we chose the same alleles used in structural analyses. We synthesized and cloned *HD1* and *HD2* into pGAD-C1 (pCH312) and pGBD-C1 (pCH478) vectors, purchased from Genewiz (Germany)[95] (Supplementary Table 3). The *Saccharomyces cerevisiae* strain PJ69-4a (CHY1268)[95] (Supplementary Table 4) was co-transformed by lithium acetate transformation using every possible combination of *HD1* and *HD2*, with each *HD* alternately serving as either bait or prey in different runs of the experiment. We selected for transformants containing both plasmids by plating on SD -leu -trp and assessed the strength of protein-protein interactions using both the *HIS3* and *lacZ* reporter genes. *HIS3* expression was determined by plating on SD -leu -trp -his + 3AT and β-galactosidase activity of transformants was determined in triplicate using o-nitrophenyl-β-galactosidase as a substrate.

## Homokaryon identification using Sanger sequencing

To discover whether homokaryotic reproduction is common, we looked for homokaryotic sporocarps in populations of *A. phalloides* introduced to North America and from European populations. We aimed to identify heterozygotic SNP sites using Sanger sequencing. We used a collection of 109 sporocarps held in the Pringle laboratory fungarium, including 40 from invasive range (15 from three sites in and around Berkeley, California, 15 from a site in New Jersey, ten from a site in New York, eight from two sites in Canada) and 69 from native range (12 from a site Montpellier, France, six from two sites in Norway, 14 from 12 sites in the UK, 13 from four sites in Austria, two from two sites in Estonia, 12 from two sites in Hungary, and two from a site in Switzerland (Supplementary Data 2). In addition, we used a collection of 30 sporocarps collected from PRNS, California in 2021 for attempting to rediscover the homokaryotic individuals (Supplementary Table 1).

After extracting DNA from sporocarps, we extracted DNA by first grinding around 10 mg of tissue in 500 µl CTAB buffer (2% cetyltrimethyl ammonium bromide, 100 mM Tris-HCl, 20 mM EDTA, and 1.4 M NaCl [pH8.0]) and incubating samples at 65 °C for 60 min. We gently mixed each solution with 500 µl 24:1 chloroform:isoamyl alcohol (Sigma-Aldrich, Missouri) for 5 min, centrifuged for 10 min at 12,000 rpm in a 5417 C centrifuge (Eppendorf, Germany), and moved supernatants to new tubes, repeating these steps twice. We then incubated supernatants with 0.6X volume of isopropanol in a −20 °C freezer overnight. We centrifuged each solution for 7 min at 12,000 rpm and removed the supernatant. The DNA pellets were washed with 1 ml ice cold 70% ethanol and centrifuged for 2 min at 12,000 rpm twice. Ethanol was discarded and the DNA pellets were dried in a Savant DNA 120 SpeedVac Concentrator (Thermo Fisher, Massachusetts) for 30 min. Finally, we eluted DNA with 50 µl water and stored the DNA extract at −20 °C.

We amplified fragments of the variable beta-flanking gene of the HD locus for the first heterozygosity screen using a C1000 Touch Thermo Cycler (Bio-Rad, California). We originally designed a primer pair to sequence a 691-mer region, but because we failed to amplify this region from many of our older specimens, we later designed a primer pair with an amplicon size of 248 nucleotides (Supplementary Table 2). For each PCR reaction, a final volume of 25 µl of reagents contained 1 µl of DNA template, 1X EconoTaq PLUS Master Mix (Lucigen, Wisconsin), 0.4 µM forward and 0.4 µM reverse primers. The PCR cycle included an initial denaturation at 95 °C for 7 min, 35 rounds of denaturing at 95 °C for 15 sec, annealing at 55 °C for 30 sec and elongation at 72 °C for 1 min, and additional 7 min of elongation at the end of the cycles. The PCR products were sequenced by Functional Biosciences (Wisconsin) on ABI 3730xl instruments (Thermo Fisher, Massachusetts).

For samples without heterozygosity at the beta-flanking gene, we designed primers for and sequenced ten additional BUSCOs located on nine different contigs (Supplementary Table 2). Each target gene fragment had at least one SNP site with estimated allele frequency

close to 0.5 in the population. We used the same amplification and sequencing methods as we used for the beta-flanking gene. Assuming the nine contigs are unlinked, and assuming each contig is itself strictly linked, we estimated the probability of misidentifying a heterozygotic sample as homozygotic to be lower than 0.002 (Supplementary Table 2).

## Statistics
The statistical tests used with yeast-two-hybrid data are described in the legend of Fig. 4.

## Reporting summary
Further information on research design is available in the Nature Portfolio Reporting Summary linked to this article.

## Data availability
The raw genomic data generated in this study have been deposited in the Sequence Read Archive (SRA) database under accession code Bioproject PRJNA565149. The assembled genome generated in this study have been deposited in GenBank under accession code JAENRT000000000.1. The variant calling data generated in this study have been deposited in the Open Science Framework repository (https://doi.org/10.17605/OSF.IO/KDE9C). The raw transcriptomic data generated in this study have been deposited in the SRA database under accession code Bioproject PRJNA689850. Genome assemblies analyzed in this study are accessible on GenBank under accession codes JNHV00000000.2, JNHY00000000.2, JMDV00000000.1 and JNHW00000000.2. Additional protein sequences analyzed in this study are accessible on GenBank under accession codes XP_001829154.1, XP_001829153.1, AAQ96344.1, AAQ96345.1, P06783.1 and AAG38536.1.

## Code availability
Scripts and additional supporting information are available on Open Science Framework (https://doi.org/10.17605/OSF.IO/BQ2RU).

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

## Acknowledgements

The authors thank Benjamin Becker, Tom Horton, Benjamin Wolfe, Cat Adams, Tom Bruns and the Bruns laboratory, Sydney Glassman, Brenda Callan, Lynne Boddy, the Pringle laboratory, Franck Richard, Milton Drott, Debbie Viess, Forrest Gander, and Ashwini Bhat for help collecting mushrooms, and fungaria for providing us with additional samples from stored collections. Field work at PRNS was conducted under permits granted to the Bruns laboratory and Y.W. (PORE-2021-SCI-0047). We thank Sarah Swanson for training in microscopy. We thank Sarah Friedrich for figure preparation. To prepare sequencing libraries, we used the University of Wisconsin-Madison Biotechnology Center's DNA Sequencing Facility (Research Resource Identifier—RRID:SCR_017759) for genomic DNA and the Biotechnology Center's Gene Expression Center Core Facility (RRID:SCR_017757) for RNA. We acknowledge various grants and fundings enabling this project, including the University of Wisconsin-Madison (A.P., Y.W.), Human Frontier Science Program grant RGP0053 (A.P.), Fulbright U.S. Scholar grant (A.P.), Mycological Society of America Graduate fellowship (Y.W.), National Institutes of Health grant T32 GM007133 (M.C.M.), Fundação para a Ciência e Tecnologia COMPETE Programme and National Funds FEDER fund PTDC/BIA-BIC/122142/2010 (S.C.G.), National Institutes of Health grant R01 AI137409 (C.M.H.).

## Author contributions

Y.W., M.C.M., H.E., J.H., J.G., H.G., C.M.H., and A.P. designed experiments. A.P., Y.W., J.G., H.E., and S.C.G. led fieldwork and Y.W., M.C.M., and H.G. performed laboratory experiments. Y.W., M.C.M., H.E., J.H., J.G., W.M., E.H., and A.P. analyzed data. A.P. and C.M.H. supervised the study. A.P. obtained funding. Y.W. wrote the initial manuscript. All authors reviewed and provided feedback towards the final manuscript.

## Competing interests

The authors declare no competing interests.
