## [Peer Review File · Nature Communications]

REVIEWER COMMENTS

Reviewer #1 (Remarks to the Author):

Dear editor,

This manuscript describes evidence for a deviating life cycle in the mushroom-forming basidiomycete fungus *Amanita phalloides*. This fungus forms ectomycorrhizas with various tree species and is the most poisonous species known. It originates in Europe but has spread to other parts of the world. In the regular life cycle of mushroom-forming basidiomycetes, sexual reproduction requires the combination of two sexually compatible haploid nuclei that first form a so-called dikaryotic mycelium (with the two nuclei remaining separate) and later, in the mushroom, have a brief diploid phase followed by meiosis. The present manuscript describes evidence that *Amanita phalloides* has two modes of reproduction. Next to the regular mode, called 'bisexuality' by the authors (but see my remark), the authors describe a different mode, called 'unisexuality' (but see my comments). In this deviating life cycle, a haploid nucleus does not require a compatible mate and can reproduce on its own. Unfortunately, this fungus cannot be cultured and so the evidence provided is based on genome data of multiple mushrooms sampled at different localities. The authors argue that this deviating life cycle can help understand why this fungus is invasive, as the ability to reproduce without the requirement to find a compatible mate, would facilitate reproduction after dispersal.

I think that this type of work, i.e. the study of natural variation in life cycles and the ecological and evolutionary implications is important, since the 'textbook' lifecycle may be less standard than we usually suppose, and the presented data illustrate this. The manuscript has potential, but I have substantial criticism on the present presentation, which needs to be addressed in a revision. My comments boil down to three general points of criticism:

1. The terminology used. Even though it may seem arbitrary what terms we use for biological phenomena, it is important to be consistent and, when using analogies, to use the right analogies. Analogies with known terms are nice, but may also be confusing, as I will outline in more detailed comments below.
2. The interpretation of the data and alternative interpretations.
3. The reference to existing literature on deviating life cycles.

Since those three points are often related, I have not categorized my points, but will discuss them one by one.

a. Unisexuality and bisexuality. Those terms suggest the presence of sexes, which is incorrect. Basidiomycete fungi are isogamous, meaning that they do not have morphologically different female and male gametes and consequently neither female and male sexes or male and female morphological structures. They do have female and male roles (with the female role defined as contributing the cytoplasm with mitochondrial genes to the spores), but those roles are not related to mating types since any mating type can play both roles. The standard life cycle is heterothallism, meaning outcrossing. Uniquely, those mushroom-forming basidiomycete fungi contain more than two, and usually up till hundreds of, mating types. For sexual compatibility, mating types must be different and given the rich variation in mating types, most combinations between random homokaryotic mycelia are sexually compatible. Mating types are sometimes equated with 'sexes' (and also here, see line 9 of the abstract and line 16 of the main text) but this is wrong since both mating partners involved in a mating provide the male and female role simultaneously.

There are exceptions to heterothallism, such as asexuality (also called parthenogenesis), and primary and secondary homothallism. To the best of my knowledge, the data presented in this paper as 'unisexuality' are consistent with both asexual reproduction, also called 'parthenogenic reproduction' (Lamouré, 1989) and primary homothallism, i.e. primary homokaryotic mycelia can fulfil the sexual life cycle on their own, without the need of a sexually compatible mate.

Genetically, asexuality can hardly be distinguished from primary homothallism, since most nuclei will be genetically identical. Either way, the results are interesting since evidence for those types of deviating lifecycles is scarce (but see below).

Not only is the biological implication of the 'sex' part in the terms unisexuality and bisexuality not warranted for the reason outlined above, also is the common-language use of the terms completely different and therefore it is very confusing for the non-expert reader ("Bisexuality is a romantic or sexual attraction or behavior toward both males and females, or to more than one gender"; "The property of being unisexual (having characteristics of a single sex).") Finding

parallels between fungal life cycles and fungal sex and better known ones is nice and can be useful too, but these parallels need to make sense biologically otherwise it will be confusing.

b. More information is needed on the standard life cycle. It would be useful if the paragraph starting with "Agaricomycete fungi are characterized by bisexual reproduction" was illustrated with a figure. As indicated in my previous comment, the term 'bisexual' is misleading, and the term 'heterothallism' should be used instead. Additionally, in line 16 the formed heterokaryon is said to be 'functionally diploid', but the authors do not mention that the two nuclei remain separate, which is very different from most other lifecycles, and has some important implications for the life cycle such as the possibility of di-mon matings. This important detail should therefore be mentioned.

c. Related to this: I believe *Amanita phalloides* does not have clamp connections? Perhaps, the authors could discuss this.

d. More information needs to be given on what is known about deviating life cycles in nature. There is abundant variation in life cycles, particularly in some genera such as *Mycena* and *Marasmius*. Next to heterothallism, asexuality (called parthenogenesis), primary and secondary homothallism and also mixtures called amphithallism are known. An old review on known deviations is Lamoure 1989 (*Cryptogamie Mycologie*, 10, pp. 41-80). More recent papers have studied *Agrocybe praecox*, where monokaryotic fruiting has been studied extensively, first by Esser and more recently by Hennicke and coworkers. Additionally, the Leslie and Lennart (1980) paper the authors refer to describes the natural occurrence of monokaryotic fruiting in *Schizophyllum commune*. I think the authors should emphasize that most of the literature predates the genomics era and therefore, that studies are urgently needed to better study the deviating life cycles, and establish the genetic details of deviating life cycles, particularly if monokaryotic fruiters reproduce asexually (parthenogenesis) or via primary homothallism. The present manuscript could be fit in that framework.

e. Related to these previous points, I think it would be useful to systematically present the data and clearly indicate what is based on 'hard evidence' and what is interpretation. What the authors have shown beyond doubt is that some of the sampled mushrooms are homokaryotic. What is not known with certainty is whether these homokaryotic mushrooms produce viable spores. There is indirect evidence that they do, since i) the same homokaryotic mushrooms of g22 are found at two widely separated localities; ii) 50% of the genome can be traced back in different heterokaryons, which by inference likely are the result of matings of different offspring of the homokaryotic mushrooms (although in theory, they could also be the result of simultaneous matings between the 'unisexual' homokaryon and other homokaryons, but this seems unlikely given the distance between samples). It will help the reader to systematically discuss and disentangle those possibilities.

f. Abstract line 11: "deadly invasive *Amanita phalloides*". Do the authors perhaps mean "invasive deadly poisonous *Amanita phalloides*"? I don't think the fungus should be considered 'deadly invasive'.

g. Abstract, line 13: "implying nuclei of homokaryotic mycelia also promote outcrossing". I would say "implying nuclei of homokaryotic mycelia also can be involved in outcrossing"

h. The sentence "However, laboratory mushrooms are typically abnormal" is strange. Do the authors perhaps refer to the previous sentence about artificially developing mushrooms from a homokaryon in the laboratory? At first, I interpreted the sentence as meaning that laboratory mushrooms typically are abnormal, and so models for what we know of mushroom lifecycles are useless, which would be strange to write. Or do the authors suggest that all life cycles deviating from heterothallism identified so far are lab artefacts? If that is the case, this is not true either (see previous point). Please clarify.

i. "Natural populations with the capacity to generate functional sporocarps both bisexually and unisexually are unknown." To the best of my knowledge this is not correct. See Lamoure 1989, Stahl et al., 1976 and other literature on species with asexuality, homothallism and heterothallism. What is true, however, is that most references are old, and therefore lack support by molecular data.

j. Page 4, lines 14-15: the term 'outgroup' is not warranted, since an outgroup is identified by the researcher beforehand, and so cannot be the result of an analysis. You could say sister group of all other taxa instead.

k. Page 4, line 5: "Many heterokaryotic individuals house either the g21 or g22 nucleus. We cannot distinguish whether these individuals are the parents or offsprings of the homokaryons...". If they are the parents, they would not have one haploid nucleus in common, since the homokaryon would be a sexual offspring of a dikaryon. If, in contrast, they were the offspring, they would have

one exact nuclear copy in common with the homokaryon. The authors could thus check, in principle, if one nucleus is in common or not, to distinguish these two possibilities. Do the data allow this? Otherwise, they may discuss this point and the data needed to distinguish, anyway.

l. Page 4, line 7: "..., nonetheless, because we identified more than one heterokaryotic individual as either the parent or offspring of both g21 and g22, and because only one individual can function as a parent of a homokaryon, we can identify other individuals as offsprings." This is a smart logic! Please use 'offspring' instead of 'offsprings' though.

m. Page 4, line 9-10: "Homokaryotic individuals appear to mate with other individuals...". Please add "thus" to indicate that this sentence logically follows from the previous, to get: "Homokaryotic individuals thus appear to mate with other individuals...".

n. In the supplemental discussion the authors discuss the possibility of 'pseudosexuality'. The authors need to explain this term. They write: "The rarity would suggest that if pseudosexuality is found in *A. phalloides*, individual g22 is the result of a single pseudosexual event." I do not get the logic. G22 can also be the result of an outcrossing sexual event, leading to a homokaryon that can form mushrooms, and perhaps spores, without a mate.

o. Page 7 line 23 – page 8 line 11: It should be noted that yeast 2-hybrid is somewhat problematic. For these assays, the absence of evidence truly doesn't mean evidence of absence. While the results they find (binding only between "outcrossing HD alleles", but not within "inbreeding alleles") are consistent with a lack of heterodimerization, it is not a strong proof. No fault of the authors, that is just a limitation of the method.

Reviewer #2 (Remarks to the Author):

This is a very interesting and thought provoking study that advances the hypothesis that selfing via unisexual reproduction has emerged in the death cap mushroom *Amanita phalloides* in association with invasion in California. Robust global sampling combined with genome analysis supports the hypothesis that individuals with a single nuclear genotype (homokaryons) are present in California, along with the more standard pattern of individuals with two nuclei of different genotypes (heterokaryons). Global sampling from many other locations revealed only heterokaryotic individuals, consistent with the model that selfing may be geographically restricted. This mushroom species is normally heterothallic, outcrossing, and thus the hypothesis is that selfing individuals arose in an otherwise out-crossing population, and that both modes of sexual reproduction are occurring in parallel and some individuals can apparently self or out-cross with an individual of compatible mating type. One general question is how well assembled are the genomes that these analyses are based on? The investigators conducted both Illumina and PacBio sequencing and thus generated hybrid assemblies. Are these complete, T2T? How many contigs, how large?

There are two key scientific questions to consider. First is how this mushroom evolved to be selfing, which in fungi is called homothallism, and whether the mating-type loci contribute. There are multiple different ways that fungi have evolved to be homothallic, and several of these involve having two compatible mating types (mating type switching, two MAT loci present in the same genome), or an auto-activated mating type system (fusions of mating type loci). The investigators explored the mating-type loci in some detail, allowing them to present a model in which bypass of the mating-type loci, one encoding homeodomain proteins and the other encoding pheromones and pheromone receptors, may have occurred. The authors find what appears to be a canonical HD mating type locus (called A in some fungi, or B in others). In the heterokaryon outcrossing state, these systems operate by forming HD1-HD2 heterodimers involving one gene product from one MAT allele with another gene product from a compatible MAT allele in the heterokaryon. In the homokaryotic state, typically the two MAT HD products do not interact, but in a selfing state this could if the two products have evolved to be self-compatible or recombination within MAT brought two compatible HD alleles together. Two hybrid tests of this did not show evidence of self-interaction, which seems to support the model. But one caveat is that the HD1-13 homeodomain did not interact with any of three HD2 partners tested for possible interaction. Thus, it is not clear if this is a technical issue of the two hybrid system, or that this MAT gene is defective or hasn't been tested with an appropriate partner thus far. The authors suggest on the basis of this evidence that the HD MAT locus is not contributing, but it may also be the case that either or both

of the homeodomain factors contribute, but not as a heterodimeric pair. Individual HD1 and HD2 factors are known to control gene expression on their own or with different partners in other fungi, and in some fungi one of the HD proteins has been lost entirely from the MAT locus and yet the other continues to play roles on its own. Resolving these possible contributions of the MAT locus would require further investigation.

The authors also suggest that they may have found the P/R mating type locus, but the region they found is atypical compared to other basidiomycete fungi and only encodes homologs of G protein coupled receptors with no linked pheromone gene (s). They mention that they found a pheromone gene elsewhere in the genome. The level of analysis presented isn't sufficient to understand what they have found here. Are these indeed pheromone receptor orthologs or homologs related to Ste3 type pheromone receptors? If so, then this should be shown by phylogenetic analysis. If not, might these receptors be related to the Cpr2 pheromone receptor that has been demonstrated to contribute to unisexual development in *Cryptococcus*, another basidiomycete species? Or alternatively, these may simply be GPCRs but not related to the specific class of pheromone receptors? What is the nature of the pheromone gene, and what is the sequence? Is there only one? Basidiomycete pheromones are all lipid modified at the C-terminal CAAX sequence, and require a STE6 pheromone exporter to have biological activity. Is there a STE6 pheromone transporter ortholog?

There is a recent example of a homothallic basidiomycete fungus in which the pheromone receptor gene and the pheromone gene lie in different regions of the genome, and the pheromone activates the receptor in an autocrine fashion to drive a selfing homothallic sexual cycle. The authors haven't mentioned or cited this finding, and it isn't clear if they have considered whether this might be part of the selfing mechanism they observe/propose for the death cap mushroom.

Obligate sexual reproduction of a homothallic fungus closely related to the *Cryptococcus* pathogenic species complex.

Passer AR, Clancey SA, Shea T, David-Palma M, Averette AF, Boekhout T, Porcel BM, Nowrousian M, Cuomo CA, Sun S, Heitman J, Coelho MA. *Elife*. 2022 Jun 17;11:e79114. doi: 10.7554/eLife.79114. PMID: 35713948 Free PMC article.

One potential limitation of the death cap mushroom system is that this species undergoes mushroom formation in nature but not as yet under lab conditions. I am also not aware that anyone has transformed it, or conducted molecular genetic studies. Thus, deleting genes to test their biological functions may not be possible, and further studies may therefore need to rely on other types of studies such as the two hybrid analysis conducted in budding yeast that is presented here. This is an approach that can be applied to test pheromone interactions with candidate pheromone receptors, such as in Tom Fowler's studies and the study cited above. Clearly further studies will be required to test if the homeodomain transcription factors, pheromone, and possible pheromone receptors are contributing to selfing.

The second key question is how the organism is undergoing selfing, and whether this might be an example of unisexual reproduction. Again the investigators consider this in light of what is known in other fungi. The investigators marshalled evidence to favor the hypothesis that the death cap mushroom is selfing via unisexual reproduction. Unisexual reproduction is a unique type of selfing in which a single individual can undergo sexual reproduction in the absence of a partner of opposite mating type, and in the absence of mating. Species that are unisexual can also still undergo out-crossing (as observed for the death cap mushroom) either by mating of cells of compatible mating type and in some cases mating between cells of the same mating type. Unisexual reproduction was discovered in the human pathogenic fungus *Cryptococcus neoformans* by Xiaorong Lin and Christina Hull when they were fellows in the Heitman lab at Duke (published as Lin et al *Nature* 2005). *Cryptococcus* is also a basidiomycete, so in the same phylum as the mushroom fungi studied here.

Sexual reproduction between partners of the same mating type in *Cryptococcus neoformans*. Lin X, Hull CM, Heitman J. *Nature*. 2005 Apr 21;434(7036):1017-21. doi: 10.1038/nature03448.

PMID: 15846346 Free article.

The authors allude to these findings, but do not actually cite this paper, which seems an oversight.

The submission also does not mention, discuss, or cite previous findings on selfing/monokaryotic fruiting/unisexual reproduction that have been reported in previous studies of two other mushroom species. These are studies from Florian Hennicke at the University of Ruhr in Bochum Germany. The species are *Cyclocybe aegerita* and *Cyclocybe parasitica*. These studies were presented recently by Florian Hennicke at the Asilomar Fungal Genetics meeting in March 2022 and the Mycological Society of America Meeting in Gainesville Florida in July 2022, and thus in addition to the publications the findings are well known in the mycology and fungal genetics community.

Volatilomes of *Cyclocybe aegerita* during different stages of monokaryotic and dikaryotic fruiting. Orban A, Hennicke F, Rühl M. *Biol Chem.* 2020 Jul 28;401(8):995-1004. doi: 10.1515/hsz-2019-0392. PMID: 32045347

The Pacific Tree-Parasitic Fungus *Cyclocybe parasitica* Exhibits Monokaryotic Fruiting, Showing Phenotypes Known from Bracket Fungi and from *Cyclocybe aegerita*. Elders H, Hennicke F. *J Fungi (Basel)*. 2021 May 19;7(5):394. doi: 10.3390/jof7050394. PMID: 34069435 Free PMC article.

Transcriptome of different fruiting stages in the cultivated mushroom *Cyclocybe aegerita* suggests a complex regulation of fruiting and reveals enzymes putatively involved in fungal oxylipin biosynthesis. Orban A, Weber A, Herzog R, Hennicke F, Rühl M. *BMC Genomics.* 2021 May 4;22(1):324. doi: 10.1186/s12864-021-07648-5. PMID: 33947322 Free PMC article.

There is clearly more work to be done in both systems but the fact that three different mushroom species have been found to undergo selfing that is thought to be occurring via unisexual reproduction in the absence of standard mating is a landmark discovery in the field. The further implication that selfing can drive geographic niche invasion sets the stage to further understand the possible broad roles of unisexual reproduction in biology, ecology, and pathogenesis.

The authors discuss in the supplemental discussion a possible alternative model for selfing in the death cap mushroom, which is pseudosexual reproduction. This was recently discovered in *Cryptococcus* and involves mating of isolates of opposite mating type, yet production of meiotic spores in which the nuclear genome is inherited entirely from one of the two parents (Yadav et al *eLife* 2021). The mechanism appears to involve the loss of one nucleus from the dikaryon, and the remaining nucleus goes on to form a diploid, undergo meiosis and produce spores. The authors mention this model and dismiss it based on the argument that the same homokaryon genotype was found at two different sampling sites locating 200 meters apart, and arguing that the species does not produce mitotic spores, and the mycelium are not thought to be this large. There are many ways in which fungi are dispersed in nature, including meiotic spores (that are produced by out-crossing, selfing, unisexual, pseudosexual reproduction) and those produced by pseudosexual reproduction might well be dispersed at a distance. Many fungi are dispersed by contact with animals, including humans, and that might also have led to dispersal of this genotype. Absent other evidence, it is very hard to see the argument that this can not be occurring via pseudosexual reproduction. The authors are welcome to favor one argument over the other, but need to sharpen the argument over what is now presented. Many basidiomycete fungi are also well known to undergo di-mon matings involving a dikaryon mating with a monokaryon, and that might also provide a route to production of homokaryotic progeny.

In summary, this is an exciting study reporting the discovery of homokaryotic mushrooms that arose via selfing, and which produce spores that have a single nucleus. This distinguishes this example of selfing from the more common form of selfing in mushrooms, which is pseudohomothallism in which two compatible nuclei are packaged into each spore. This finding alone is novel. The fact that it may also be linked with invasive potential is thought-provoking. The analysis of the whole genome allows specific models to be considered and tested. The authors have marshalled their evidence to make the case that this may represent an example of unisexual reproduction, and it may be. To know how this is controlled at the molecular level, and to what extent the MAT loci are contributing or are bypassed would require further investigation.

Specific/minor comment. The legends for Figure 3 and some of the extended data figures could be more explicit in describing the structures seen in the SEM imaging.

RESPONSE TO REVIEWERS' COMMENTS

Original letter and reviews in black font, our responses in blue.

Reviewer #1 (Remarks to the Author):

Dear editor,

This manuscript describes evidence for a deviating life cycle in the mushroom-forming basidiomycete fungus *Amanita phalloides*. This fungus forms ectomycorrhizas with various tree species and is the most poisonous species known. It originates in Europe but has spread to other parts of the world. In the regular life cycle of mushroom-forming basidiomycetes, sexual reproduction requires the combination of two sexually compatible haploid nuclei that first form a so-called dikaryotic mycelium (with the two nuclei remaining separate) and later, in the mushroom, have a brief diploid phase followed by meiosis. The present manuscript describes evidence that *Amanita phalloides* has two modes of reproduction. Next to the regular mode, called 'bisexuality' by the authors (but see my remark), the authors describe a different mode, called 'unisexuality' (but see my comments). In this deviating life cycle, a haploid nucleus does not require a compatible mate and can reproduce on its own. Unfortunately, this fungus cannot be cultured and so the evidence provided is based on genome data of multiple mushrooms sampled at different localities. The authors argue that this deviating life cycle can help understand why this fungus is invasive, as the ability to reproduce without the requirement to find a compatible mate, would facilitate reproduction after dispersal.

I think that this type of work, i.e. the study of natural variation in life cycles and the ecological and evolutionary implications is important, since the 'textbook' lifecycle may be less standard than we usually suppose, and the presented data illustrate this.

We appreciate the thoughtful (and very accurate!) summary of our findings and we are glad the reviewer finds our work important. Thank you for the kind words and thoughtful critiques on our manuscript.

The manuscript has potential, but I have substantial criticism on the present presentation, which needs to be addressed in a revision. My comments boil down to three general points of criticism:

1. The terminology used. Even though it may seem arbitrary what terms we use for biological phenomena, it is important to be consistent and, when using analogies, to use the right analogies. Analogies with known terms are nice, but may also be confusing, as I will outline in more detailed comments below.

We also think terminology is important.

2. The interpretation of the data and alternative interpretations.

3. The reference to existing literature on deviating life cycles.

Since those three points are often related, I have not categorized my points, but will discuss them one by one.

We will respond to each point below.

a. Unisexuality and bisexuality. Those terms suggest the presence of sexes, which is incorrect. Basidiomycete fungi are isogamous, meaning that they do not have morphologically different female and male gametes and consequently neither female and male sexes or male and female morphological structures.

While we have enormous regard for the reviewer's opinion, we also respectfully stand by our use of the emerging terminology of unisexuality and bisexuality, originally developed by Joe Heitman and his lab for use to describe the α/α mating in *Cryptococcus neoformans*, and subsequently adopted by scientists including Brenda Wingfield and her lab for use with *Huntiaella moniliformis* (Wilson, et al. 2015), and thereafter extended to other ascomycetes (Wilson, et al. 2021).

Wilson, Andrea M., et al. "Unisexual reproduction in *Huntiaella moniliformis*." *Fungal Genetics and Biology* 80 (2015): 1-9.

Wilson, Andi M., et al. "Doing it alone: unisexual reproduction in filamentous ascomycete fungi." *Fungal Biology Reviews* 35 (2021): 1-13.

They do have female and male roles (with the female role defined as contributing the cytoplasm with mitochondrial genes to the spores),

Not every mycologist would agree with this statement, and the statement may be especially tricky when we consider basidiomycete fungi. Even though in uniparental mitochondrial inheritance with unilateral nuclear migration, one mycelium would have the "male" role (contributing nuclei) and the other have the "female" role (receiving the nucleus, but not sending a nucleus back; the mycelium receiving the nucleus also houses the mitochondria), many species of fungi seem to inherit mitochondria from both "parents", while the nuclei migrate in both directions. In this second system of mating, both mycelia contribute mitochondria and nuclei, but the cells of derived dikaryons are not heteroplasmic (i.e. no cells have a mixture of different types of mitochondria) and it would be hard to infer male and female roles (some cells have the mitochondria of one "parent", some have cells of the other, and the mycelium is a mitochondrial mosaic). There are also no male and female roles in biparental mitochondrial inheritance (also a feature of some fungi), since both mycelia are contributing mitochondria into the same cell. In our recently published study, we found that

A. phalloides' mitochondria are uniparentally inherited, but we have little evidence to support if the nuclei are unilaterally or bilaterally migrated. (Wang et al. 2023).

Wang, Yen-Wen, Holly Elmore, and Anne Pringle. "Uniparental Inheritance and Recombination as Strategies to Avoid Competition and Combat Muller's Ratchet among Mitochondria in Natural Populations of the Fungus *Amanita phalloides*." *Journal of Fungi* 9.4 (2023): 476.

but those roles are not related to mating types since any mating type can play both roles. The standard life cycle is heterothallism, meaning outcrossing. Uniquely, those mushroom-forming basidiomycete fungi contain more than two, and usually up till hundreds of, mating types. For sexual compatibility, mating types must be different and given the rich variation in mating types, most combinations between random homokaryotic mycelia are sexually compatible. Mating types are sometimes equated with 'sexes' (and also here, see line 9 of the abstract and line 16 of the main text) but this is wrong since both mating partners involved in a mating provide the male and female role simultaneously.

Thank you for the suggestion. We have removed "sexes" from page 1 L9 and page 2 L16 for the purpose of accuracy.

There are exceptions to heterothallism, such as asexuality (also called parthenogenesis), and primary and secondary homothallism. To the best of my knowledge, the data presented in this paper as 'unisexual' are consistent with both asexual reproduction, also called 'parthenogenic reproduction' (Lamoure, 1989) and primary homothallism, i.e. primary homokaryotic mycelia can fulfil the sexual life cycle on their own, without the need of a sexually compatible mate.

We note the long, often tortured history associated with definitions of asexuality. Our favorite reference on this is Graham Bell's book "The Masterpiece of Nature: The Evolution and Genetics of Sexuality" (University of California Press). The book includes many chapters on how biologists through time (and currently) have defined and debated words like asexuality and parthenogenesis. Often, "sex" is equated with mixis (very roughly, meiosis, although we note some forms of asexuality involved meiosis) and "not sex" is equivalent to "amixis" (and according to Bell there are (at least) six different kinds of amixis, and according to him all of them are mitotic).

But for asexuality involving meiosis, see also: Levitis, Daniel A., Kolea Zimmerman, and Anne Pringle. "Is meiosis a fundamental cause of inviability among sexual and asexual plants and animals?." *Proceedings of the Royal Society B: Biological Sciences* 284.1860 (2017): 20170939.

At any rate, as far as we can interpret, Lamoure's definition for parthenogenic reproduction also invokes meiosis, and follows Kühner's findings of a two-spored form of *Mycena* (Petersen, 1995). While these fungi normally produce four spores per basidium, each of which will germinate as a homokaryotic mycelium, the spores of the two-spored form germinate into heterokaryotic mycelium. This finding is clearly not what we have discovered in the homokaryotic *A. phalloides*.

If we take that asexuality only encompasses mitotic reproduction, which is sometimes called "haploid parthenogenesis" (described by Stahl and Esser, 1976), then we should consider if what we have discovered involves only mitosis. We believe our system is not mitotic. The mitotic parthenogenetic monokaryotic sporocarps described by Stahl and Esser have two spores per basidium. Each spore receives one of the two mitotic products. Whereas the basidia in typical sporocarps bear four spores and each spore receives one of the four meiotic products. Although the number of basidiospores in *A. phalloides* is not consistent, the close-to-mature basidia of unisexual individuals hold four nuclei. With this evidence in mind, we conclude meiosis is most likely involved in what we have discovered. We have added a short discussion on this topic in the new section "Comparing *A. phalloides* to other basidiomycete fungi" in this revision.

"Primary homothallism" also has multiple definitions. Originally it referred to homothallism in which only one copy of a genome was involved (a contrast to pseudohomothallism). More recently, the definition of primary homothallism has changed to require the fungus to have two compatible MAT loci (using a bipolar ancestral state as an example) in the same genome (something that is mostly caused by gene duplications; Wilson et al. 2015). Our system is also not primary homothallism, if it is defined as it is currently, because we find no signatures of duplications of the HD locus, nor are the two products of the *HD* genes compatible. While our observation does accord with the older definition of primary homothallism, our data are a better fit for the emerging terminology of unisexuality, as defined here by our second reviewer, they write:

"Unisexual reproduction is a unique type of selfing in which a single individual can undergo sexual reproduction in the absence of a partner of opposite mating type, and in the absence of mating. Species that are unisexual can also still undergo out-crossing (as observed for the death cap mushroom) either by mating of cells of compatible mating type and in some cases mating between cells of the same mating type."

We agree with this reviewer's description of unisexuality and it is the definition we are using in our manuscript.

Petersen, Ronald H. "There's more to a mushroom than meets the eye: mating studies in the Agaricales." *Mycologia* 87.1 (1995): 1-17.

Stahl, Ulf, and Karl Esser. "Genetics of fruit body production in higher basidiomycetes: I. Monokaryotic fruiting and its correlation with dikaryotic fruiting in *Polyporus ciliatus*." *Molecular and General Genetics MGG* 148 (1976): 183-197.

Wilson, Andrea M., et al. "Homothallism: an umbrella term for describing diverse sexual behaviours." *IMA fungus* 6.1 (2015): 207-214.

Genetically, asexuality can hardly be distinguished from primary homothallism, since most nuclei will be genetically identical. Either way, the results are interesting since evidence for those types of deviating lifecycles is scarce (but see below).

Not only is the biological implication of the 'sex' part in the terms unisexuality and bisexuality not warranted for the reason outlined above, also is the common-language use of the terms completely different and therefore it is very confusing for the non-expert reader ("Bisexuality is a romantic or sexual attraction or behavior toward both males and females, or to more than one gender"; "The property of being unisexual (having characteristics of a single sex).") Finding parallels between fungal life cycles and fungal sex and better known ones is nice and can be useful too, but these parallels need to make sense biologically otherwise it will be confusing.

We thank the reviewer for pointing out potential confusions and hope for more discussions in future. In the context of the current manuscript, we have broken down our next discussion into three points: 1. The use of the word bisexuality; 2. Isogamy and bisexuality; 3. Use of homothallism and heterothallism. (See also our previous discussions of some of these points, above.)

1. Use of bisexuality

We agree that "bisexuality", like the word sex itself, has different meanings in everyday life and popular culture *versus* science, and that using it alone without context may generate confusions. However, we provide a clear definition for bisexuality in the manuscript (at page 2 L15–L17, we say "To complete the life cycle, two haploid mycelia of different mating types fuse and form a functionally diploid (heterokaryotic) mycelium..."), and we believe readers will not be confused when contrasting bisexuality with unisexuality as we use it for *A. phalloides*. In addition, the use of bisexuality has been accepted and used in the same manner by other fungal biologists. The terminology was first used in *Cryptococcus neoformans* by Joe Heitman and his lab to describe a/α mating. It was subsequently adopted by other groups studying the same genus (Gyawali et al, 2017), as well as groups that focus on other fungi (Wilson et al. 2018).

2. Isogamy and bisexuality

Sexual reproduction in eukaryotes can be roughly categorized into isogamy, anisogamy and oogamy. For our purposes here we will not consider oogamy. As the reviewer has described, basidiomycetes may or may not be isogamous or

of different sizes fuse/mate, or perhaps mating involves di/mon mating, or... there are other possibilities). In part for this reason, and as we have written above, morphologically defined sexual differentiation makes little sense; in our view basidiomycetes do not grow as morphological "females" or "males". And so in our manuscript, we are not discussing our findings with morphological sexes in mind. Rather, we discuss sexuality by considering mating types involved in the act of sexual reproduction (the generation of mushrooms), and very much adhere to concepts defined by a different classic literature, e.g. by John Raper (1966). And sexual reproduction involving a single mating type qualifies that individual as unisexual; likewise, sexual reproduction involving two mating types qualifies that individual as bisexual. As the reviewer has pointed out, the former manuscript did use the word "sexes", and like we mentioned above, we have removed those two incidents.

3. Use of heterothallism

We had many, many discussions about whether or not to use the word heterothallism instead of bisexuality to describe the canonical death cap genomes/individuals. We chose not to use heterothallism because the direct opposing term, homothallism, is not a precise term to describe the non-canonical death cap genomes/individuals. The issue with homothallism is that formally, it describes sexual reproduction occurring from a single mycelium (homo=single and thallus=body); the mycelium or "body" generates "sexual" spores without another partner. But that formal definition is an umbrella term because the phenomenon of generating spores from a single body encompasses many different mechanisms, including unisexuality, primary homothallism (sensu Wilson et al. 2015), mating type switching, and pseudohomothallism. While unisexuality involves one mating type, both mating type switching and pseudohomothallism should be considered bisexual because there are two different mating types involved. Our use of the word unisexual highlights the uniqueness of our finding (one body makes basidiospores on its own and only one mating type is involved), clearly distinguishes the unisexual from the bisexual *A. phalloides*, and distinguishes the dynamics of reproduction in *A. phalloides* from other, more well-known mechanisms (e.g., mating type switching).

Fu, Ci, et al. "Unisexual versus bisexual mating in *Cryptococcus neoformans*: Consequences and biological impacts." *Fungal Genetics and Biology* 78 (2015): 65-75.

Gyawali, Rachana, et al. "Pheromone independent unisexual development in *Cryptococcus neoformans*." *PLoS genetics* 13.5 (2017): e1006772.

Raper, John R. "Genetics of sexuality in higher fungi." *Genetics of sexuality in higher fungi*. (1966).

Wilson, Andrea M., et al. "Homothallism: an umbrella term for describing diverse sexual behaviours." *IMA fungus* 6.1 (2015): 207-214.

Wilson, Andi M., et al. "Pheromone expression reveals putative mechanism of unisexuality in a saprobic ascomycete fungus." *PLoS One* 13.3 (2018): e0192517.

b. More information is needed on the standard life cycle. It would be useful if the paragraph starting with "Agaricomycete fungi are characterized by bisexual reproduction" was illustrated with a figure. As indicated in my previous comment, the term 'bisexual' is misleading, and the term 'heterothallism' should be used instead. Additionally, in line 16 the formed heterokaryon is said to be 'functionally diploid', but the authors do not mention that the two nuclei remain separate, which is very different from most other lifecycles, and has some important implications for the life cycle such as the possibility of di-mon matings. This important detail should therefore be mentioned.

As suggested, we have created a new figure (included below), and in the figure we have made the phenomenon of a mycelium made up of cells with two distinct/separate nuclei very clear.

We also added a short note about the two nuclei remaining separate on page 2 L17–18.

Fig. 5. Current model of the life cycle of *Amanita phalloides*, illustrating a bisexual reproductive cycle (left) and a unisexual reproductive cycle (right).

c. Related to this: I believe *Amanita phalloides* does not have clamp connections? Perhaps, the authors could discuss this.

That is correct. They don't have clamp connections as shown in Extended Data Fig. 4. We have included a short description of our observations on page 5 L19 and in the figure legend.

d. More information needs to be given on what is known about deviating life cycles in nature. There is abundant variation in life cycles, particularly in some genera such as *Mycena* and *Marasmius*. Next to heterothallism, asexuality (called parthenogenesis), primary and secondary homothallism and also mixtures called amphithallism are known. An old review on known deviations is Lamoure 1989 (*Cryptogamie Mycologie*, 10, pp. 41-80). More recent papers have studied *Agrocybe praecox*, where monokaryotic fruiting has been studied extensively, first by Esser and more recently by Hennicke and coworkers. Additionally, the

Leslie and Lennart (1980) paper the authors refer to describes the natural occurrence of monokaryotic fruiting in *Schizophyllum commune*.

Like the reviewer, we have enormous respect for the literature on deviating life cycles, and perhaps especially for an older literature (including the Lamoure 1989 paper, but also papers by Lorna Casselton) which bravely described fungal sex without the advantages of the tools we use today. We have added a new section to describe the diversity of fungal sexual reproduction at page 10 L1–21.

About Leslie and Lennart's 1980 paper: Leslie did not observe monokaryotic sporocarps directly in nature (personal communication), instead, using spores collected from the wild, they attempted monokaryotic fruiting in the lab – and some cultures were successfully induced. We have clarified at page 3 L2–3.

I think the authors should emphasize that most of the literature predates the genomics era and therefore, that studies are urgently needed to better study the deviating life cycles, and establish the genetic details of deviating life cycles, particularly if monokaryotic fruiters reproduce asexually (parthenogenesis) or via primary homothallism. The present manuscript could be fit in that framework.

We agree our manuscript could fit in the described framework! As mentioned above, we have added a paragraph to page 10 L1–21 to place our work in a broader context. But much of what the reviewer suggests is beyond the scope of our already packed paper. It would be fantastic to develop a review to both talk about older literature and talk about the emerging use of modern molecular tools, and especially as they can be applied to understanding the ecology and evolution of fungi in nature. We would gladly take up the task of writing an exciting, forward-looking review of fungal sex in nature, and why it matters, and what textbooks say as opposed to what is most likely happening, what we can discover with new tools... but we think that would have to be a separate paper.

e. Related to these previous points, I think it would be useful to systematically present the data and clearly indicate what is based on 'hard evidence' and what is interpretation. What the authors have shown beyond doubt is that some of the sampled mushrooms are homokaryotic. What is not known with certainty is whether these homokaryotic mushrooms produce viable spores. There is indirect evidence that they do, since i) the same homokaryotic mushrooms of g22 are found at two widely separated localities;

This is very strong evidence for reproduction by viable spores: The alternative hypothesis is mycelial growth by a single individual across more than 200 meters, downhill and uphill and across a water course. We have worked with these populations for more than 20 years and have a strong sense of the ecology of *A. phalloides*; we are writing a paper with data directly measuring the size of bodies in nature including measurements from this study (please let

us know if you would like to see a copy and we will gladly send the current draft). No body is as large as 200 meters and most bodies are very small (less than 5 m across).

ii) 50% of the genome can be traced back in different heterokaryons, which by inference likely are the result of matings of different offspring of the homokaryotic mushrooms (although in theory, they could also be the result of simultaneous matings between the 'unisexual' homokaryon and other homokaryons, but this seems unlikely given the distance between samples).

Again, this is very strong evidence for the viability of unisexual spores. Other possibilities are improbable at best.

It will help the reader to systematically discuss and disentangle those possibilities.

We take the point that many readers will ask the same kinds of questions and have added a short discussion of viability at page 9 L17–L18.

f. Abstract line 11: "deadly invasive *Amanita phalloides*". Do the authors perhaps mean "invasive deadly poisonous *Amanita phalloides*"? I don't think the fungus should be considered 'deadly invasive'.

Thank you for the suggestion, we have revised the language to "invasive and deadly *Amanita phalloides*".

g. Abstract, line 13: "implying nuclei of homokaryotic mycelia also promote outcrossing". I would say "implying nuclei of homokaryotic mycelia also can be involved in outcrossing"

We have revised the language to "implying nuclei of homokaryotic mycelia are also involved in outcrossing".

h. The sentence "However, laboratory mushrooms are typically abnormal" is strange. Do the authors perhaps refer to the previous sentence about artificially developing mushrooms from a homokaryon in the laboratory? At first, I interpreted the sentence as meaning that laboratory mushrooms typically are abnormal, and so models for what we know of mushroom lifecycles are useless, which would be strange to write. Or do the authors suggest that all life cycles deviating from heterothallism identified so far are lab artefacts? If that is the case, this is not true either (see previous point). Please clarify.

Thank you for pointing out the confusion. What we meant is that the structures of mushrooms developed from haploid mycelia grown in the lab are often abnormal (e.g. Stahl and Esser, 1976). We have revised the line to "However, mushrooms forced from haploid mycelia grown in laboratory environments are typically abnormal".

i. "Natural populations with the capacity to generate functional sporocarps both bisexually and unisexually are unknown." To the best of my knowledge this is not correct. See Lamoure 1989, Stahl et al., 1976 and other literature on species with asexuality, homothallism and heterothallism. What is true, however, is that most references are old, and therefore lack support by molecular data.

Thank you for pointing out the confusion. Our original description is inaccurate. What we were trying to express is that populations that generate functional sporocarps both bisexually and unisexually in natural environments are unknown. We have revised the line accordingly (page 3 L2–3).

j. Page 4, lines 14-15: the term 'outgroup' is not warranted, since an outgroup is identified by the researcher beforehand, and so cannot be the result of an analysis. You could say sister group of all other taxa instead.

We have revised the lines as suggested, and also revised the legends of Extended Data Fig. 2.

k. Page 4, line 5: "Many heterokaryotic individuals house either the g21 or g22 nucleus. We cannot distinguish whether these individuals are the parents or offsprings of the homokaryons...". If they are the parents, they would not have one haploid nucleus in common, since the homokaryon would be a sexual offspring of a dikaryon. If, in contrast, they were the offspring, they would have one exact nuclear copy in common with the homokaryon. The authors could thus check, in principle, if one nucleus is in common or not, to distinguish these two possibilities. Do the data allow this? Otherwise, they may discuss this point and the data needed to distinguish, anyway.

This is a great idea! Unfortunately, we only sequenced the mushrooms with an Illumina platform, and the use of that platform doesn't allow us to phase the whole chromosomes of the mushrooms. Therefore, it is not feasible for us to test if one nucleus of the heterokaryons is exactly the same as the nucleus of the homokaryons. However, in a different paper that is more generally about the evolution of mitochondria we have analyzed the mitochondrial genomes of the homokaryons (see Wang et al. 2023). All but one of the heterokaryons that are either parents or offsprings of the g22 homokaryon have the same mitochondrial genome as g22. Since homokaryons must have the same mitochondrial genome as their parent, we can be sure that the homokaryon with a different mitochondrial genome (g7) is an offspring. This piece of evidence pushes the age of g22 back to nearly 30 years and provides additional support for spores as viable. The paper describing the population genomics of mitochondria was very recently published (as we note above as well). We decided to not include any mitochondrial findings in this paper to avoid replicating results between manuscripts.

Wang, Y.-W.; Elmore, H.; Pringle, A. Uniparental inheritance and recombination as strategies to avoid competition and combat Muller's ratchet among mitochondria in natural populations of the fungus *Amanita phalloides*. *J. Fungi* 2023, 9, 476.
<https://doi.org/10.3390/jof9040476>

l. Page 4, line 7: "..., nonetheless, because we identified more than one heterokaryotic individual as either the parent or offspring of both g21 and g22, and because only one individual can function as a parent of a homokaryon, we can identify other individuals as offsprings." This is a smart logic! Please use 'offspring' instead of 'offsprings' though.

m. Page 4, line 9-10: "Homokaryotic individuals appear to mate with other individuals...". Please add "thus" to indicate that this sentence logically follows from the previous, to get: "Homokaryotic individuals thus appear to mate with other individuals...".

Thank you for the encouragement! We have revised the two lines accordingly.

n. In the supplemental discussion the authors discuss the possibility of 'pseudosexuality'. The authors need to explain this term. They write: "The rarity would suggest that if pseudosexuality is found in *A. phalloides*, individual g22 is the result of a single pseudosexual event." I do not get the logic. G22 can also be the result of an outcrossing sexual event, leading to a homokaryon that can form mushrooms, and perhaps spores, without a mate.

We agree the supplementary discussion for pseudosexuality was not well written. We have entirely rewritten it using simpler logic and we hope it is now very clear, please see Supplementary Discussion—"The natural history of the homokaryon g22 does not support an hypothesis of pseudosexuality".

o. Page 7 line 23 – page 8 line 11: It should be noted that yeast 2-hybrid is somewhat problematic. For these assays, the absence of evidence truly doesn't mean evidence of absence. While the results they find (binding only between "outcrossing HD alleles", but not within "inbreeding alleles") are consistent with a lack of heterodimerization, it is not a strong proof. No fault of the authors, that is just a limitation of the method.

Thank you for pointing out this caveat. We are aware of it and therefore we wrote "The heterodimerization of HD proteins from the homokaryotic MAT allele is 'unlikely' to enable sporocarp development" and "Our finding 'suggests' other mechanisms besides heterodimerization enable the development of homokaryotic mushrooms." For clarity, in this revision we also changed "the same MAT allele would not interact" to "the same MAT allele would not produce any significant signal of an interaction." And "The HD1 and HD2 proteins from the homokaryotic sporocarp also did not interact with each other or with themselves under the conditions tested" to "The HD1 and HD2 proteins from the

homokaryotic sporocarp also did not produce any significant signal of an interaction with each other or with themselves under the conditions tested". (page 8 L12 and L15)

Reviewer #2 (Remarks to the Author):

This is a very interesting and thought provoking study that advances the hypothesis that selfing via unisexual reproduction has emerged in the death cap mushroom *Amanita phalloides* in association with invasion in California. Robust global sampling combined with genome analysis supports the hypothesis that individuals with a single nuclear genotype (homokaryons) are present in California, along with the more standard pattern of individuals with two nuclei of different genotypes (heterokaryons). Global sampling from many other locations revealed only heterokaryotic individuals, consistent with the model that selfing may be geographically restricted. This mushroom species is normally heterothallic, outcrossing, and thus the hypothesis is that selfing individuals arose in an otherwise outcrossing population, and that both modes of sexual reproduction are occurring in parallel and some individuals can apparently self or out-cross with an individual of compatible mating type. One general question is how well assembled are the genomes that these analyses are based on? The investigators conducted both Illumina and PacBio sequencing and thus generated hybrid assemblies. Are these complete, T2T? How many contigs, how large?

We thank the reviewer for their kind words and succinct summary of the importance of our findings. Our hybrid assembly for the reference genome is not a complete T2T assembly. In total there are 605 scaffolds, ranging from 913 to 1,833,728 bp. N50 and NG50 are 320 kbp and 184 kbp respectively. This information is provided at Supplementary Methods– "Reference genome assembly".

There are two key scientific questions to consider. First is how this mushroom evolved to be selfing, which in fungi is called homothallism, and whether the mating-type loci contribute. There are multiple different ways that fungi have evolved to be homothallic, and several of these involve having two compatible mating types (mating type switching, two MAT loci present in the same genome), or an auto-activated mating type system (fusions of mating type loci). The investigators explored the mating-type loci in some detail, allowing them to present a model in which bypass of the mating-type loci, one encoding homeodomain proteins and the other encoding pheromones and pheromone receptors, may have occurred. The authors find what appears to be a canonical HD mating type locus (called A in some fungi, or B in others). In the heterokaryon outcrossing state, these systems operate by forming HD1-HD2 heterodimers involving one gene product from one MAT allele with another gene product from a compatible MAT allele in the heterokaryon. In the homokaryotic state, typically the two MAT HD products do not interact, but in a selfing state this could if the two products have evolved to be self-compatible or recombination within MAT brought two compatible HD alleles together. Two hybrid tests of this did not show

evidence of self-interaction, which seems to support the model. But one caveat is that the HD1-13 homeodomain did not interact with any of three HD2 partners tested for possible interaction. Thus, it is not clear if this is a technical issue of the two hybrid system, or that this MAT gene is defective or hasn't been tested with an appropriate partner thus far. The authors suggest on the basis of this evidence that the HD MAT locus is not contributing, but it may also be the case that either or both of the homeodomain factors contribute, but not as a heterodimeric pair. Individual HD1 and HD2 factors are known to control gene expression on their own or with different partners in other fungi, and in some fungi one of the HD proteins has been lost entirely from the MAT locus and yet the other continues to play roles on its own. Resolving these possible contributions of the MAT locus would require further investigation.

We thank the reviewer for their succinct and thoughtful summary of our findings! Indeed, we are also intrigued by the behavior of HD1-13. Our interpretation is that only one pair of the HD1 and HD2 is needed to form a heterodimer and to enable mating. We are aware the HD1 and/or HD2 may still contribute to sporocarp development without heterodimerization. Therefore, at page 8 L16–19 we wrote “Thus, the heterodimerization of HD proteins from the homokaryotic MAT allele is unlikely to enable sporocarp development. Our finding suggests other mechanisms besides heterodimerization enable the development of homokaryotic mushrooms.”

We also agree further investigation is needed to parse out these hypotheses, and we will likely need additional genetic manipulations to distinguish among them.

The authors also suggest that they may have found the P/R mating type locus, but the region they found is atypical compared to other basidiomycete fungi and only encodes homologs of G protein coupled receptors with no linked pheromone gene (s). They mention that they found a pheromone gene elsewhere in the genome. The level of analysis presented isn't sufficient to understand what they have found here. Are these indeed pheromone receptor orthologs or homologs related to Ste3 type pheromone receptors? If so, then this should be shown by phylogenetic analysis. If not, might these receptors be related to the Cpr2 pheromone receptor that has been demonstrated to contribute to unisexual development in *Cryptococcus*, another basidiomycete species? Or alternatively, these may simply be GPCRs but not related to the specific class of pheromone receptors?

These are not easy questions to answer, and particularly as regards *A. phalloides*, which is not even remotely a model system (and neither is the genus *Amanita*, even though it includes many of the world's most famous mushrooms. We are among the first to use genetics to describe its mating behaviors). At any rate, while Ste3-like genes are well understood in *Cryptococcus*, where there are only two Ste3-like genes (each with a “mating-type factor pheromone receptor activity” GO term), among the Agaricales, Ste-3-like genes have undergone a huge expansion. However, in *A. phalloides*, there are only two genes

annotated as "mating-type factor pheromone receptor activity", and these two genes are also the only two hits when we use the STE3 genes in *Coprinopsis cinerea* in a BLAST search with a default E-value cutoff of 10. To understand the orthology of these two genes (from *A. phalloides*) we reconstructed a phylogeny and discovered that the two STE3-like genes are not orthologous to any of the mating-type determining genes among Agaricales (Extended Data Fig. 7). However, Ap.00g075660 is orthologous to the Cpr2 pheromone receptor in *Cryptococcus neoformans*, while Ap.00g075660 is not (Additional supporting information https://osf.io/bq2ru/?view_only=9f17ee8694ff49858e0fa0b75a1854dc). Because of the complications in reconstructing phylogenies for these highly divergent proteins (in most cases, allele divergence predates speciation), we do not wish to overinterpret our results.

What is the nature of the pheromone gene, and what is the sequence? Is there only one? Basidiomycete pheromones are all lipid modified at the C-terminal CAAX sequence, and require a STE6 pheromone exporter to have biological activity. Is there a STE6 pheromone transporter ortholog?

We actually found two pheromone genes. The sequences of the pheromone genes are provided in Supplementary figure 1 (and included down below). Both pheromone genes encode the canonical -CaaX motif and we have added a description of the genes to the revised manuscript (page 6 L13–14).

Thank you for pointing out STE6. We did not analyze this gene specifically. For this revision, we attempted to identify STE6 in *A. phalloides* by BLAST. We identified 27 genes. Although one gene (Ap.00g014230) has a significant higher identity to the STE6 in *Cryptococcus neoformans* (45.83%) than the others and was functionally annotated as a type-1 ABC transporter. However, it is hard to interpret the function of this gene without genetic tools. For example, pheromone secretion of some Agaricales may be independent of STE6 (Fowler et al., 1999). Because we cannot make strong conclusions based on this analysis, we decided not to include the analysis in our revision.

Supplementary Fig. 1. Putative pheromone precursors. The two pheromones both show the presence of the -CxxA motifs, characteristic of fungal pheromones. Neither gene was found near a PR gene.

Fowler, Thomas J., et al. "Multiple sex pheromones and receptors of a mushroom-producing fungus elicit mating in yeast." *Molecular Biology of the Cell* 10.8 (1999): 2559-2572.

There is a recent example of a homothallic basidiomycete fungus in which the pheromone receptor gene and the pheromone gene lie in different regions of the genome, and the pheromone activates the receptor in an autocrine fashion to drive a selfing homothallic sexual cycle. The authors haven't mentioned or cited this finding, and it isn't clear if they have considered whether this might be part of the selfing mechanism they observe/propose for the death cap mushroom.

Obligate sexual reproduction of a homothallic fungus closely related to the *Cryptococcus* pathogenic species complex.

Passer AR, Clancey SA, Shea T, David-Palma M, Averette AF, Boekhout T, Porcel BM, Nowrousian M, Cuomo CA, Sun S, Heitman J, Coelho MA. *Elife*. 2022 Jun 17;11:e79114. doi: 10.7554/eLife.79114. PMID: 35713948 Free PMC article.

We did consider autocrine signaling as potentially involved in *A. phalloides* mating, and we know this kind of signaling was also discussed by David-Palma et al. (2016). We chose not to discuss bipolarity in the submitted version of the manuscript, but we agree that it would benefit readers to include some mention of it. We have added a short discussion of bipolarity at page 6 L21–24.

David-Palma, Marcia, Jose Paulo Sampaio, and Paula Goncalves. "Genetic dissection of sexual reproduction in a primary homothallic basidiomycete." *PLoS Genetics* 12.6 (2016): e1006110.

One potential limitation of the death cap mushroom system is that this species undergoes mushroom formation in nature but not as yet under lab conditions. I am also not aware that anyone has transformed it, or conducted molecular genetic studies. Thus, deleting genes to test their biological functions may not be possible, and further studies may therefore need to rely on other types of studies such as the two hybrid analysis conducted in budding yeast that is presented here. This is an approach that can be applied to test pheromone interactions with candidate pheromone receptors, such as in Tom Fowler's studies and the study cited above. Clearly further studies will be required to test if the homeodomain transcription factors, pheromone, and possible pheromone receptors are contributing to selfing.

Death caps are terrible lab organisms. We agree that further investigation of pheromone and pheromone receptors will be essential for understanding the complete machinery of mating in *A. phalloides*. While additional experiments are beyond the scope of our current study, and our current manuscript is already packed enough. In future we aim to combine our ecological focus and field work with exactly the kind of laboratory manipulations suggested by the reviewer. To understand how *A. phalloides* and other fungi mate in nature, we will need a synthesis of skills, and will need to go back-and-forth between populations in

nature and laboratory models, and creatively use experimentally tractable fungi to probe what's going on with unculturable fungi.

The second key question is how the organism is undergoing selfing, and whether this might be an example of unisexual reproduction. Again the investigators consider this in light of what is known in other fungi. The investigators marshalled evidence to favor the hypothesis that the death cap mushroom is selfing via unisexual reproduction. Unisexual reproduction is a unique type of selfing in which a single individual can undergo sexual reproduction in the absence of a partner of opposite mating type, and in the absence of mating. Species that are unisexual can also still undergo out-crossing (as observed for the death cap mushroom) either by mating of cells of compatible mating type and in some cases mating between cells of the same mating type.

Unisexual reproduction was discovered in the human pathogenic fungus *Cryptococcus neoformans* by Xiaorong Lin and Christina Hull when they were fellows in the Heitman lab at Duke (published as Lin et al Nature 2005). *Cryptococcus* is also a basidiomycete, so in the same phylum as the mushroom fungi studied here.

Sexual reproduction between partners of the same mating type in *Cryptococcus neoformans*.

Lin X, Hull CM, Heitman J. Nature. 2005 Apr 21;434(7036):1017-21. doi: 10.1038/nature03448. PMID: 15846346 Free article.

The authors allude to these findings, but do not actually cite this paper, which seems an oversight.

Thank you for pointing out this missing citation. We have added it to page 10 L15 in our revision.

The submission also does not mention, discuss, or cite previous findings on selfing/monokaryotic fruiting/unisexual reproduction that have been reported in previous studies of two other mushroom species. These are studies from Florian Henricke at the University of Ruhr in Bochum Germany. The species are *Cyclocybe aegerita* and *Cyclocybe parasitica*. These studies were presented recently by Florian Henricke at the Asilomar Fungal Genetics meeting in March 2022 and the Mycological Society of America Meeting in Gainesville Florida in July 2022, and thus in addition to the publications the findings are well known in the mycology and fungal genetics community.

Volatilomes of *Cyclocybe aegerita* during different stages of monokaryotic and dikaryotic fruiting. Orban A, Henricke F, Rühl M. Biol Chem. 2020 Jul 28;401(8):995-1004. doi: 10.1515/hsz-2019-0392. PMID: 32045347

The Pacific Tree-Parasitic Fungus *Cyclocybe parasitica* Exhibits Monokaryotic Fruiting, Showing Phenotypes Known from Bracket Fungi and from *Cyclocybe aegerita*. Elders H,

Hennicke F. J Fungi (Basel). 2021 May 19;7(5):394. doi: 10.3390/jof7050394. PMID: 34069435
Free PMC article.

Transcriptome of different fruiting stages in the cultivated mushroom *Cyclocybe aegerita* suggests a complex regulation of fruiting and reveals enzymes putatively involved in fungal oxylipin biosynthesis. Orban A, Weber A, Herzog R, Hennicke F, Rühl M. BMC Genomics. 2021 May 4;22(1):324. doi: 10.1186/s12864-021-07648-5. PMID: 33947322 Free PMC article.

We thank the reviewer for pointing us to these missing citations. The study in *Cyclocybe* is indeed highly relevant to our study. We have included a discussion about these two species and cited the papers as suggested at page 10 L17.

There is clearly more work to be done in both systems but the fact that three different mushroom species have been found to undergo selfing that is thought to be occurring via unisexual reproduction in the absence of standard mating is a landmark discovery in the field. The further implication that selfing can drive geographic niche invasion sets the stage to further understand the possible broad roles of unisexual reproduction in biology, ecology, and pathogenesis.

We agree: The growing number of observations of fungal species (also *Volvariella volvacea*) as being able to reproduce unisexually as well as bisexually is changing our understanding of sexual reproduction in basidiomycetes. Much exciting follow-up research will be required to further our knowledge of unisexual reproduction, including but not limited to the potential link between unisexual reproduction and invasiveness, and the genetic drivers of unisexual reproduction.

The authors discuss in the supplemental discussion a possible alternative model for selfing in the death cap mushroom, which is pseudosexual reproduction. This was recently discovered in *Cryptococcus* and involves mating of isolates of opposite mating type, yet production of meiotic spores in which the nuclear genome is inherited entirely from one of the two parents (Yadav et al eLife 2021). The mechanism appears to involve the loss of one nucleus from the dikaryon, and the remaining nucleus goes on to form a diploid, undergo meiosis and produce spores. The authors mention this model and dismiss it based on the argument that the same homokaryon genotype was found at two different sampling sites locating 200 meters apart, and arguing that the species does not produce mitotic spores, and the mycelium are not thought to be this large. There are many ways in which fungi are dispersed in nature, including meiotic spores (that are produced by out-crossing, selfing, unisexual, pseudosexual reproduction) and those produced by pseudosexual reproduction might well be dispersed at a distance. Many fungi are dispersed by contact with animals, including humans, and that might also have led to dispersal of this genotype.

Quick note: We have evidence against animal/human-mediated dispersal of *A. phalloides*; populations at sites visited frequently by humans are still genetically distinct (there is strong isolation-by-distance) which we would not find if humans were frequently carrying *A. phalloides* around the Californian sites where we work.

Absent other evidence, it is very hard to see the argument that this can not be occurring via pseudosexual reproduction. The authors are welcome to favor one argument over the other, but need to sharpen the argument over what is now presented. Many basidiomycete fungi are also well known to undergo di-mon matings involving a dikaryon mating with a monokaryon, and that might also provide a route to production of homokaryotic progeny.

We agree the supplementary discussion for pseudosexuality was not well written. We have entirely rewritten it using simpler logic and we hope it is now very clear, please see Supplementary Discussion—"The natural history of the homokaryon g22 does not support an hypothesis of pseudosexuality".

In summary, this is an exciting study reporting the discovery of homokaryotic mushrooms that arose via selfing, and which produce spores that have a single nucleus. This distinguishes this example of selfing from the more common form of selfing in mushrooms, which is pseudohomothallism in which two compatible nuclei are packaged into each spore. This finding alone is novel. The fact that it may also be linked with invasive potential is thought-provoking. The analysis of the whole genome allows specific models to be considered and tested. The authors have marshalled their evidence to make the case that this may represent an example of unisexual reproduction, and it may be. To know how this is controlled at the molecular level, and to what extent the MAT loci are contributing or are bypassed would require further investigation.

We thank the reviewer for their encouragement. We too believe that our findings provide a novel understanding of fungal sexual reproduction. Because *A. phalloides* is unculturable, we will need to turn to other systems to describe the biology of *A. phalloides* further and understand its unisexuality at a molecular level.

Specific/minor comment. The legends for Figure 3 and some of the extended data figures could be more explicit in describing the structures seen in the SEM imaging.

We have added arrowheads and arrows to SEM images to indicate immature spores and basidia.

REVIEWER COMMENTS

Reviewer #1 (Remarks to the Author):

The authors have done a great job in revising the manuscript, and also in responding in detail to my suggestions. I still do not agree in all respects, particularly on some of the choices of the terminology (although some additions and small modifications certainly helped), but feel that this should not further delay the acceptance. I only have a few rather minor suggestions and comments, which I added as track changes or as comments in the annotated version that I attach. As I said, I am happy with the detailed response to my comments, but have a few "responses to the response", which I attach as a separate file. I wish the authors good luck with the finalisation of their manuscript, and thank them for the nice review process.

Reviewer #2 (Remarks to the Author):

This is an interesting and thought provoking study that advances the hypothesis that selfing via unisexual reproduction has emerged in the death cap mushroom *Amanita phalloides* in association with invasion in California. The manuscript has been revised in response to two reviews. As outlined in further detail below, too much of the response to reviews only appears to be in the response to reviewers comments, and not included as revisions in the paper or supplemental materials. Additional analyses that are critical for the arguments presented need to be documented such that readers, like the reviewers, do not continue to have the same questions about what the authors found.

The authors have spent considerable time and effort responding to the reviews, which is very much appreciated. At this stage, based on the data presented, and how it is presented, it is fair to say that they have advanced a very provocative model/hypothesis, but that the evidence falls short of proving the molecular nature for the observation of selfing. The analysis is observational and not at the current state of development mechanistic. It is also fair to say that this is going to require a great deal of additional experimentation that are beyond the state of the art, and out of the scope of this first report. The current state of the genomes is a highly fragmented assembly (605 scaffolds) and improving this will likely require additional primary sequence data (PacBio HiFi, Nanopore) and better assemblies.

The authors provide in the response to reviews a great deal of additional information about the two pheromone receptor candidates they have identified, one of which they state is an ortholog of the *Cpr2* receptor from *Cryptococcus*. This information was not included in the revised manuscript or the extensive supplemental information. The authors included a link to additional supporting information in the response to reviews:

However, Ap.00g075660 is orthologous to the *Cpr2* pheromone receptor in *Cryptococcus neoformans*, while Ap.00g075660 is not (Additional supporting information https://osf.io/bq2ru/?view_only=9f17ee8694ff49858e0fa0b75a1854dc).

This reviewer visited this supporting information link, and there is a very large amount of data and files there, which isn't adequately labelled or organized in any way that would enable a reader or reviewer to navigate what is located there. The supplemental section submitted with the revised manuscript provides a general description of how the authors searched for pheromone receptors and pheromone genes, but does not include the description, analysis, presentation of what they found. This is summarized here in the response to reviews, but the primary data on which these conclusions are based needs to be part of the published manuscript. First, this should include the phylogenetic analysis of the two candidate pheromone receptors, one of which the authors assert is an ortholog of *Cpr2*. Second, the authors included a figure labelled Supplemental Fig. 1 showing the analysis of the two pheromone genes they have identified. I could not find this figure in the revised paper or the extended data figures. This figure needs to be included somewhere in the paper.

The authors mention in response to reviews that another species, *Volvarella volvacea*, has been

shown to reproduce unisexually. Has this been published? If so, the authors should mention and cite the publication(s) in the introduction or discussion. What is the evidence that this species is selfing?

The authors state in the abstract that development of homokaryotic mushrooms appears to bypass mating type gene control, and this hypothesis may well turn out to be correct. But even if the pheromone and pheromone receptor genes/locus are found to not control mating type, they may still be contributing to promote development, either by virtue of a receptor being constitutively active, or if this species is producing a pheromone that acts on its own pheromone receptor to drive autocrine signaling. This would appear to be different from models in which the role of the mating type components have been bypassed. As currently written, this alternative hypothesis does not seem to be considered.

In the introduction, two reviews written by the same graduate student from the Wingfield lab are cited (references 10, 11) as the background for unisexual reproduction in fungi. It would be appropriate to cite in this context the landmark paper from Xiaorong Lin and Christina Hull that first provided definitive proof that fungi can reproduce unisexually (Lin et al Nature 2005). There might also be others reviews on this general subject that might appropriately be cited as well.

There is clearly more work to be done, but the fact that three (or even four) different mushroom species have been found to undergo selfing that is thought to be occurring via unisexual reproduction in the absence of standard mating is a landmark discovery in the field. The further implication that selfing can drive geographic niche invasion sets the stage to further understand the possible broader roles in biology, ecology, and pathogenesis.

In summary, this is an exciting study reporting the discovery of homokaryotic mushrooms that arose via selfing, and which produce spores that have a single nucleus. This distinguishes this example of selfing from the more common form of selfing in mushrooms, which is pseudohomothallism in which two compatible nuclei are packaged into each spore. This finding alone is novel. The fact that it may also be linked with invasive potential is thought-provoking. The analysis of the whole genome allows specific models to be considered and tested. The authors have marshalled their evidence to make the case that this may represent an example of unisexual reproduction, and it may be. To know how this is controlled at the molecular level, and to what extent the MAT loci are contributing or are bypassed would require further investigation. It is this reviewers opinion that when the points raised above have been addressed, that this will make an outstanding contribution to the field and will be a foundation to build upon in future studies to determine the detailed mechanisms.

Review addendum:

The authors are quite right that Tom Fowler reported previously that when basidiomycete pheromones from *S. commune* are expressed heterologously in *S. cerevisiae*, some pheromones appear to be highly dependent on export via Ste6, others are in part dependent upon Ste6 for export, and others appear not to require Ste6 for export.

This suggests that some of these lipid modified pheromones may be exported via other MDR related pumps when expressed in the model yeast *S. cerevisiae*.

In contrast, in the basidiomycete *C. neoformans* pheromone export is largely dependent upon the Ste6 ortholog identified in that basidiomycete species.

See: Hsueh, Yen-Ping, and Wei-Chiang Shen. "A homolog of Ste6, the a-factor transporter in *Saccharomyces cerevisiae*, is required for mating but not for monokaryotic fruiting in *Cryptococcus neoformans*." *Eukaryotic cell* 4.1 (2005): 147-155.

Thus, it may be that the STE6 ortholog found in any given basidiomycete species may have evolved to export the cognate pheromones from that species whereas when expressed heterologously in a very different ascomycete yeast, the transporter may not have evolved to transport the heterologous ligands.

RESPONSE TO REVIEWERS' COMMENTS

Original review in black; response in blue

Reviewer #1 (Remarks to the Author):

The authors have done a great job in revising the manuscript, and also in responding in detail to my suggestions. I still do not agree in all respects, particularly on some of the choices of the terminology (although some additions and small modifications certainly helped), but feel that this should not further delay the acceptance. I only have a few rather minor suggestions and comments, which I added as track changes or as comments in the annotated version that I attach. As I said, I am happy with the detailed response to my comments, but have a few "responses to the response", which I attach as a separate file. I wish the authors good luck with the finalisation of their manuscript, and thank them for the nice review process.

We thank the reviewer for the encouragement and have also appreciated the nice review process! We have copied the reviewer's thoughts here in green and respond to their thoughts in orange.

From "responses to the response":

This is a misunderstanding: also in 'regular' matings between two homokaryons, i.e. with migration of nuclei in both directions, mitochondrial inheritance generally is uniparental. The reason is that the mitochondria do not migrate (in contrast to the nuclei). This implies that mushrooms formed on the left side of a mating, will transmit mitochondria from the left side, and mushrooms formed on the right side mitochondria from the right side. Mitochondrial transmission is thus 'doubly uniparental' (except for the small chance that a mushroom is formed at the zone where the two homokaryons met, where cells could be heteroplasmic). So mitochondrial transmission is uniparental at the level of spores (or mushrooms). At the level of the dikaryotic mycelium mitochondrial transmission is biparental. All this is explained in Aanen et al (2004, Proc.R.Soc.B).

Thank you for these thoughts, fungal mitochondria are so interesting and also understudied in nature. We know something about how the mitochondria within *A. phalloides* are inherited and recently published our findings in Journal of Fungi. We often found multiple mushrooms belonging to the same genet, in other words, a mycelium growing underground would produce multiple sporocarps aboveground. But we did not find any genet that produces mushrooms with different mitochondria in different mushrooms, suggesting death caps do not normally inherit mitochondria from both parents.

Wang, Yen-Wen, Holly Elmore, and Anne Pringle. "Uniparental Inheritance and Recombination as Strategies to Avoid Competition and Combat Muller's Ratchet among

Mitochondria in Natural Populations of the Fungus *Amanita phalloides*." *Journal of Fungi* 9.4 (2023): 476.

This addition is very helpful indeed!

Thank you for the kind words!

Oogamy is a form of anisogamy, i.e. one where one of the gametes is immobile.

Of course the reviewer is right – in retrospect, we shouldn't have distinguished between anisogamy and oogamy.

As I wrote in a comment in the manuscript, I found it confusing that the basidiospores contain two nuclei. Even though this is correct, I would suggest to simplify this and draw only a single nucleus (while adding a note in the figure legend to explain that this is a simplification). Since the two nuclei are mitotically derived they are genetically identical and so it is not really important for your story, hence my suggestion.

Thank you for carefully thinking through the figure. Instead of simplifying the life cycle, we have clarified why there are two nuclei per cell by adding text to the legend. We suspect having two mitotically derived nuclei in each spore is a key to understanding the biology of invasive death caps and so we believe it is important to illustrate the detail in the figure. It's also true our Fig. 3f and Fig. S4b,c show images of basidiospores with multiple nuclei, and we seek to avoid conflict among our figures. But by adding text to the legend, we hope we have made the figure quite clear.

From "annotations to the word document":

I do not think this addition is necessary for the abstract, as it just introduces jargon without adding anything to the previous words.

We agree and have edited accordingly.

I would put it less strongly ("may have facilitated")

With the history of invasion biology in mind, we prefer to keep our original language.

I saw that you draw the basidiospores in the schematic lifecycle in Figure 5 with two nuclei. While this is correct, I suggest to simplify this as the two nuclei are genetically identical in any case, and it may be confusing. Instead you could add a remark in the legend on this, that you simplified it in the figure...

Please see our comments above.

Reviewer #2 (Remarks to the Author):

This is an interesting and thought provoking study that advances the hypothesis that selfing via unisexual reproduction has emerged in the death cap mushroom *Amanita phalloides* in association with invasion in California. The manuscript has been revised in response to two reviews. As outlined in further detail below, too much of the response to reviews only appears to be in the response to reviewers comments, and not included as revisions in the paper or supplemental materials. Additional analyses that are critical for the arguments presented need to be documented such that readers, like the reviewers, do not continue to have the same questions about what the authors found.

Thank you for the work you have once again put into our manuscript. We seek to make all analyses easily accessible to readers, and towards that aim and with the reviewer's comments in mind we have included additional analyses in the main text and restructured the supplementary files to resolve potential issues. Changes include:

1. The merging of Extended Data with Supplementary Information (now the figure for pheromones is located with other supplementary figures).
2. The reorganization of Supplementary Data (mainly changing supplementary notes to supplementary results and moved it to the beginning of the file)
3. The addition of table of contents for Supplementary Information.
4. The addition of more discussions (page 6 L19–20 and supplementary discussions: "Interspecies phylogeny of PR genes", and "Exploration of pheromone transporter") and a supplementary figure (Fig. S9) of STE3 *Cr. neoformans*.

The authors have spent considerable time and effort responding to the reviews, which is very much appreciated. At this stage, based on the data presented, and how it is presented, it is fair to say that they have advanced a very provocative model/hypothesis, but that the evidence falls short of proving the molecular nature for the observation of selfing. The analysis is observational and not at the current state of development mechanistic.

We agree. As ecologists who've long envied what can be done with model fungi in laboratory settings, we are glad to have been able to use genomes to make the discovery described in our manuscript, and like the reviewer, think the discovery of unisexuality in the context of an invasion is important. But elucidating the specific molecular mechanisms involved will require deep creativity and a great deal more than genomes.

It is also fair to say that this is going to require a great deal of additional experimentation that are beyond the state of the art, and out of the scope of this first report. The current state of the genomes is a highly fragmented assembly (605 scaffolds) and improving this

will likely require additional primary sequence data (PacBio HiFi, Nanopore) and better assemblies.

Yes. With these data in hand, we hope to justify additional funding for different and better genomes.

The authors provide in the response to reviews a great deal of additional information about the two pheromone receptor candidates they have identified, one of which they state is an ortholog of the Cpr2 receptor from *Cryptococcus*. This information was not included in the revised manuscript or the extensive supplemental information. The authors included a link to additional supporting information in the response to reviews:

However, Ap.00g075660 is orthologous to the Cpr2 pheromone receptor in *Cryptococcus neoformans*, while Ap.00g075660 is not (Additional supporting information https://osf.io/bq2ru/?view_only=9f17ee8694ff49858e0fa0b75a1854dc).

This reviewer visited this supporting information link, and there is a very large amount of data and files there, which isn't adequately labelled or organized in any way that would enable a reader or reviewer to navigate what is located there.

The supporting information is available through the Open Science Framework, and a key to understanding the organization of the files is the "README" file, which should be read first. We have gone back and changed the titles of our various headings within the OSF project to make more clear what the files are/what they are about, and have changed the title of the "README" file to "README_FIRST". In that file, we have elaborated on what each file is/how each file connects to the manuscript and we also corrected a few inaccuracies.

Also, we have noticed a mistake in our previous response. It should be: "Ap.00g075660 is orthologous to the Cpr2 pheromone receptor in *Cryptococcus neoformans*, while 'Ap.00g075670' is not".

The supplemental section submitted with the revised manuscript provides a general description of how the authors searched for pheromone receptors and pheromone genes, but does not include the description, analysis, presentation of what they found. This is summarized here in the response to reviews, but the primary data on which these conclusions are based needs to be part of the published manuscript. First, this should include the phylogenetic analysis of the two candidate pheromone receptors, one of which the authors assert is an ortholog of Cpr2.

The primary data involve a phylogeny (a species/gene reconciled tree) of 32 species and 174 sequences. It is unwieldy as either a main or supplemental figure and needs to be kept in the OSF project. However, in addition to labelling it more carefully within the project, we have more thoroughly described it in the file README_FIRST.

We have also created a new figure specifically for the supplement that is more readable: in this figure, we only include the two PRs in *A. phalloides* and the two PRs in *C. neoformans*. We have added this new figure to the supplement at the same point where we discuss the original species/gene reconciliation.

Second, the authors included a figure labelled Supplemental Fig. 1 showing the analysis of the two pheromone genes they have identified. I could not find this figure in the revised paper or the extended data figures. This figure needs to be included somewhere in the paper.

We apologize for the confusion. In previous versions of our manuscript, the supplementary figures were either in extended data or supplementary information, because we transferred the manuscript from another journal, and we had formatted the manuscript according to the protocols of that other journal. Note: this figure is now labeled as Supplementary Fig. 7.

The authors mention in response to reviews that another species, *Volvariella volvacea*, has been shown to reproduce unisexually. Has this been published? If so, the authors should mention and cite the publication(s) in the introduction or discussion. What is the evidence that this species is selfing?

Yes, this is published and we discussed and cited it at "results and discussion" (page 10 L21). The observation is based on single spore isolations and PCR.

Chen, B. *et al.* Fruiting body formation in *Volvariella volvacea* can occur independently of its MAT-A-controlled bipolar mating system, enabling homothallic and heterothallic life cycles. *G3-Genes Genom. Genet.* (2016) doi:10.1534/g3.116.030700.

The authors state in the abstract that development of homokaryotic mushrooms appears to bypass mating type gene control, and this hypothesis may well turn out to be correct. But even if the pheromone and pheromone receptor genes/locus are found to not control mating type, they may still be contributing to promote development, either by virtue of a receptor being constitutively active, or if this species is producing a pheromone that acts on its own pheromone receptor to drive autocrine signaling. This would appear to be different from models in which the role of the mating type components have been bypassed. As currently written, this alternative hypothesis does not seem to be considered.

We have a short discussion of this alternative hypothesis from page 6 L23 to page 7 L2. To be more explicit, we added "Autoactivated PRs from self-secreted pheromones, constitutively active PRs, as well as a bypass in the molecular pathway of sexual development are each alternative hypotheses for the irrelevance of PRs in mating type determination in *A. phalloides*." Afterwards.

In the introduction, two reviews written by the same graduate student from the Wingfield lab are cited (references 10, 11) as the background for unisexual reproduction in fungi. It would be appropriate to cite in this context the landmark paper from Xiaorong Lin and Christina Hull that first provided definitive proof that fungi can reproduce unisexually (Lin et al Nature 2005). There might also be others reviews on this general subject that might appropriately be cited as well.

We have replaced one of the Wingfield lab citations with a citation to the paper from Xiarong Lin and a book chapter by Roach *et al*.

Roach, K. C., Feretzaki, M., Sun, S., & Heitman, J. (2014). Unisexual reproduction. *Advances in genetics*, 85, 255-305.

There is clearly more work to be done, but the fact that three (or even four) different mushroom species have been found to undergo selfing that is thought to be occurring via unisexual reproduction in the absence of standard mating is a landmark discovery in the field. The further implication that selfing can drive geographic niche invasion sets the stage to further understand the possible broader roles in biology, ecology, and pathogenesis.

Thank you for your appreciation of our research!

In summary, this is an exciting study reporting the discovery of homokaryotic mushrooms that arose via selfing, and which produce spores that have a single nucleus. This distinguishes this example of selfing from the more common form of selfing in mushrooms, which is pseudohomothallism in which two compatible nuclei are packaged into each spore. This finding alone is novel. The fact that it may also be linked with invasive potential is thought-provoking. The analysis of the whole genome allows specific models to be considered and tested. The authors have marshalled their evidence to make the case that this may represent an example of unisexual reproduction, and it may be. To know how this is controlled at the molecular level, and to what extent the MAT loci are contributing or are bypassed would require further investigation. It is this reviewers opinion that when the points raised above have been addressed, that this will make an outstanding contribution to the field and will be a foundation to build upon in future studies to determine the detailed mechanisms.

Thank you!

Review addendum:

The authors are quite right that Tom Fowler reported previously that when basidiomycete

pheromones from *S. commune* are expressed heterologously in *S. cerevisiae*, some pheromones appear to be highly dependent on export via Ste6, others are in part dependent upon Ste6 for export, and others appear not to require Ste6 for export. This suggests that some of these lipid modified pheromones may be exported via other MDR related pumps when expressed in the model yeast *S. cerevisiae*.

In contrast, in the basidiomycete *C. neoformans* pheromone export is largely dependent upon the Ste6 ortholog identified in that basidiomycete species.

See: Hsueh, Yen-Ping, and Wei-Chiang Shen. "A homolog of Ste6, the a-factor transporter in *Saccharomyces cerevisiae*, is required for mating but not for monokaryotic fruiting in *Cryptococcus neoformans*." *Eukaryotic cell* 4.1 (2005): 147-155.

Thus, it may be that the STE6 ortholog found in any given basidiomycete species may have evolved to export the cognate pheromones from that species whereas when expressed heterologously in a very different ascomycete yeast, the transporter may not have evolved to transport the heterologous ligands.

This is so interesting – and perhaps illustrates the complications of drawing conclusions about function when a gene from one phylum of fungi is placed within a species from a different phylum (although heterologous gene expression is a tool we all use regularly...).